# Capillary trapping of various nanomaterials on additively manufactured scaffolds for 3D micro-/nanofabrication

Xianglong Lyu[1,2], Zhiqiang Zheng[1], Anitha Shiva[1], Mertcan Han[1,2], Cem Balda Dayan[3], Mingchao Zhang[1] ✉ & Metin Sitti[1,2,4] ✉

High-precision additive manufacturing technologies, such as two-photon polymerization, are mainly limited to photo-curable polymers and currently lacks the possibility to produce multimaterial components. Herein, we report a physically bottom-up assembly strategy that leverages capillary force to trap various nanomaterials and assemble them onto three-dimensional (3D) microscaffolds. This capillary-trapping strategy enables precise and uniform assembly of nanomaterials into versatile 3D microstructures with high uniformity and mass loading. Our approach applies to diverse materials irrespective of their physiochemical properties, including polymers, metals, metal oxides, and others. It can integrate at least four different material types into a single 3D microstructure in a sequential, layer-by-layer manner, opening immense possibilities for tailored functionalities on demand. Furthermore, the 3D microscaffolds are removable, facilitating the creation of pure material-based 3D microstructures. This universal 3D micro-/nanofabrication technique with various nanomaterials enables the creation of advanced miniature devices with potential applications in multifunctional microrobots and smart micromachines.

Three-dimensional (3D) micro-/nanofabrication using various materials has emerged as a compelling technology[1–3] with considerable potentials in numerous fields, including microrobotics[4–7], micro-metamaterials[8,9], micro-electrodes/electronics[10], and micro-/nanooptics[11]. Among these techniques, two-photon polymerization (2PP) stands out as a cutting-edge method for 3D micro-/nanofabrication. It enables the creation of intricate, free-form 3D micro-/nanostructures with unparalleled design flexibility, surpassing the capabilities of traditional manufacturing[2,3]. In 2PP, two-photon absorption (2PA) at the focal point of a femtosecond laser induces the polymerization and solidification of monomers with a resolution up to 100 nm[12]. However, this process is limited by the specific requirements of 2PA, which restricts the range of usable monomers[13–15]. Consequently, the materials are confined to a narrow spectrum polymeric materials, such as commercially available inert polymers[4,16,17] and a limited variety of custom-made functional hydrogels[5,6,18,19].

Two strategies have been reported for 3D printing of non-polymeric materials, such as metals[20], quantum dots[21–23], metal oxides[24,25], and ceramics[26–28] at the micro-/nanoscale. One strategy relies on the direct laser writing of advanced photoresists, which can either include designing cross-linkable precursors for targeted materials[20,21,25,27–29] or creating blends by physically dispersing nanomaterials into accessible monomers[26,30]. After printing, the desired materials are either chemically bonded to other heterogeneous materials or physically embedded within a polymer matrix. To achieve pure 3D structures of the targeted material, post-treatments, such as pyrolysis and etching, are necessary to remove the bonding

[1]Physical Intelligence Department, Max Planck Institute for Intelligent Systems, Stuttgart, Germany. [2]Institute for Biomedical Engineering, ETH Zürich, Zürich, Switzerland. [3]Robotic Materials Department, Max Planck Institute for Intelligent Systems, Stuttgart, Germany. [4]School of Medicine and College of Engineering, Koç University, Istanbul, Turkey. ✉e-mail: zhangmc16@tsinghua.org.cn; sitti@is.mpg.de

materials[20,26,29]. For example, a colloidal nanocrystal photoresist was developed by grafting cross-linkable ligands onto the nanocrystal surface, enabling the direct laser writing of various 3D architectures with a high nanocrystal mass content of ≈90%[29]. However, this method is generally limited to specific chemistries, such as the surface ligands on colloidal nanocrystals (including C−H, Zn-S bonds[21,29]) and cross-linkable metal coordination compounds as certain metal precursors, hindering its broader applicability to other different materials. Besides, for photoresists composed of physically blended nanomaterials within accessible monomers, strict criteria of the nanomaterials need to be met, such as the size, mass loading, and transparency of the blended nanomaterials[26,30].

Another strategy to fabricate non-polymeric 3D micro-/nanostructures involves template-assisted fabrication, which uses 3D-printed polymer scaffolds as templates for the deposition/adsorption of desired materials[31]. Techniques, such as chemical coating[32], physical vapor deposition[8], atomic layer deposition[33–35], and weak interaction-based adsorption (electrostatic force[36] or hydrogen bonding[37]), have been used to deposit various inorganic materials onto these templates. These methods have produced inorganic material-coated structures with high resolution (≈100 nm) and decent mechanical strength. However, they still face the challenges, such as the limited applicability to other materials, low mass loading of the desired materials, and the difficulties of integrating multiple materials. Detailed comparisons of the printing capabilities of existing approaches are provided in Supplementary Table 1. Despite these advancements, developing a versatile 3D micro-/nanofabrication strategy that broadly accommodates a wide spectrum of materials (both polymeric and non-polymeric materials), achieves high material content, and facilitates the on-demand integration of multiple materials remains a challenge.

Here, we propose a general strategy for the 3D micro-/nanofabrication of various materials, utilizing a non-specific physical interaction, i.e., capillary forces[38–40], to trap nanomaterial dispersions and assemble the nanomaterials onto 3D 2PP-printed microscaffolds. The 2PP-printed scaffolds possess a strong pinning ability at the liquid/solid interface, effectively trapping the solution within and directing the steady assembly of nanomaterials onto the microscaffolds during the evaporation process. To ensure uniform and abundant nanomaterial deposition, we minimize the surface charge of the nanomaterials through salting out effect and reduce their electrostatic repulsion. By trapping and confining various nanomaterial dispersed solution inside the microscaffolds, we can realize the creation of 3D micro-/nanostructures from a diverse array of materials, including polymer, quantum dots, metal, metal oxide, alloy, diamond, and up-conversion nanomaterials. Additionally, we show that the polymer scaffolds can be removed, yielding pure-material 3D microstructures. Moreover, our method allows for easy integration of multiple materials, especially sequential integration of multiple materials into a single microscaffold. This capability has great potential for fabrication of heterogeneous, multifunctional micro-/nanodevices tailored for specific, synergetic, and on-demand functionalities.

## Results

### Capillary trapping-enabled 3D micro-/nanofabrication

The creation of material-nonspecific 3D micro-/nanostructures involves a unique "capillary trapping" strategy, where nanomaterials are trapped and assembled onto microscaffolds through the capillary force. This process is outlined in Fig. 1a−c and includes several steps: (I) Modifying the silicon substrate with silane molecules; (II) Processing polymer microscaffolds (made from IPS photoresist polymer) on the silicon substrate through the 2PP process; (III) Repeatedly immersing and retracting the substrate in nanomaterial dispersions. The nanomaterials are gradually accumulated and assembled onto the microscaffolds after the trapped solution within them evaporates in each

immersion-retraction cycle (Fig. 1c). Particularly, the capillary force, a prevalent non-specific physical interaction on solid-liquid-gas three-phase interfaces, plays a vital role during the enrichment and deposition process of nanomaterials at the micro-/nanoscale. It effectively traps the solution (≈0.12 μL for each microscaffolds; for details, see Methods) within the microscaffolds in each immersion-retraction cycle. The surface treatment of the substrate reduces its affinity for the solution, while the untreated microscaffolds exhibit a higher affinity. Differential affinity between the substrate and microscaffolds results in a directional flow of the dispersion liquid to the microscaffolds. This process facilitates the concentrated deposition of nanomaterials onto the microscaffolds, while it remains clear on the substrate, as the scanning electron microscope (SEM) image shows in Fig. 1c.

The microscaffolds on the substrate are crucial for precisely capturing the solution and guiding the assembly and deposition of nanomaterials within these structures. Without these scaffolds, nanomaterials accumulating at the solid-liquid-air interfaces form flat 'coffee-ring' patterns[41,42]. This pattern results from capillary flows, which move nanomaterials to the edge to compensate for solution loss during lateral evaporation process, leading to the accumulation of nanomaterials at the interface (Supplementary Fig. 1). In contrast, within the microscaffolds, the 3D microstructures provide a stronger pinning ability, ensuring the solution remains within the microscaffolds. As the evaporation occurs, the bulk solution on the substrate, carrying nanomaterials, tends to directionally flow into these 3D microscaffolds to replenish the evaporated solution due to its higher affinity for the 3D microscaffolds (Fig. 1c and Supplementary Movie 1). This leads to the first accumulation phase, creating a 3D liquid-air interface confined by the microscaffolds. This accumulation increases the nanomaterial concentration, facilitating a high nanomaterial loading onto the 3D microscaffolds during subsequent evaporation stages within the microscaffolds.

Sequentially, as evaporation continues within the 3D microscaffolds, local contact lines form along each building block, leading to a second phase of nanomaterial accumulation (the second accumulation) at the meniscus of each contact line. The steady accumulation (Supplementary Fig. 2) indicates that the interparticle van der Waals forces are robust enough to counteract the capillary forces during ongoing deposition cycles. Furthermore, we immersed the as-assembled scaffolds in a pure solvent to assess if the capillary forces during evaporation could dislodge nanoparticles from the scaffolds (Supplementary Fig. 3). Even after 600 evaporation cycles, there was no noticeable change in the deposition morphology on the scaffolds. These findings lead us to conclude that although van der Waals forces are generally weak, they are adequately strong to overcome the capillary forces, particularly given that nanoparticles possess high surface areas, enabling a high interparticle adhesion strength. To facilitate a dense and uniform deposition, a small amount of salt is added to reduce the electrostatic interactions among the nanomaterials. This salting-out effect neutralizes the surface charge of the nanomaterials, diminishing their electrostatic repulsion[43–45] (Supplementary Fig. 4). As the solvent evaporates, the salt concentration increases, further accelerating the coagulation of the nanomaterials[46,47].

Our strategy assembles Au NPs onto 3D microscaffolds, as demonstrated in the SEM images (Fig. 1d−f) and the elemental mapping from energy-dispersive X-ray spectroscopy (EDS) data (Fig. 1g). These images show that Au NPs are uniformly and densely packed onto each building block of the 3D microscaffold. This technique enables us to create diverse Au NP-assembled 3D microstructures with a high structural integrity. Examples include a micro "bird's nest" (Fig. 1h, i) and a micro-metastructure (Fig. 1j, k). As this strategy leverages the prevalent physical interactions−capillary forces (a noticeable interaction at small scales[38]), to guide the assembly of nanomaterials, and van der Waals forces to stabilize the assembled nanomaterial layers−it can assemble nanomaterials onto micro-/nanostructures with submicron

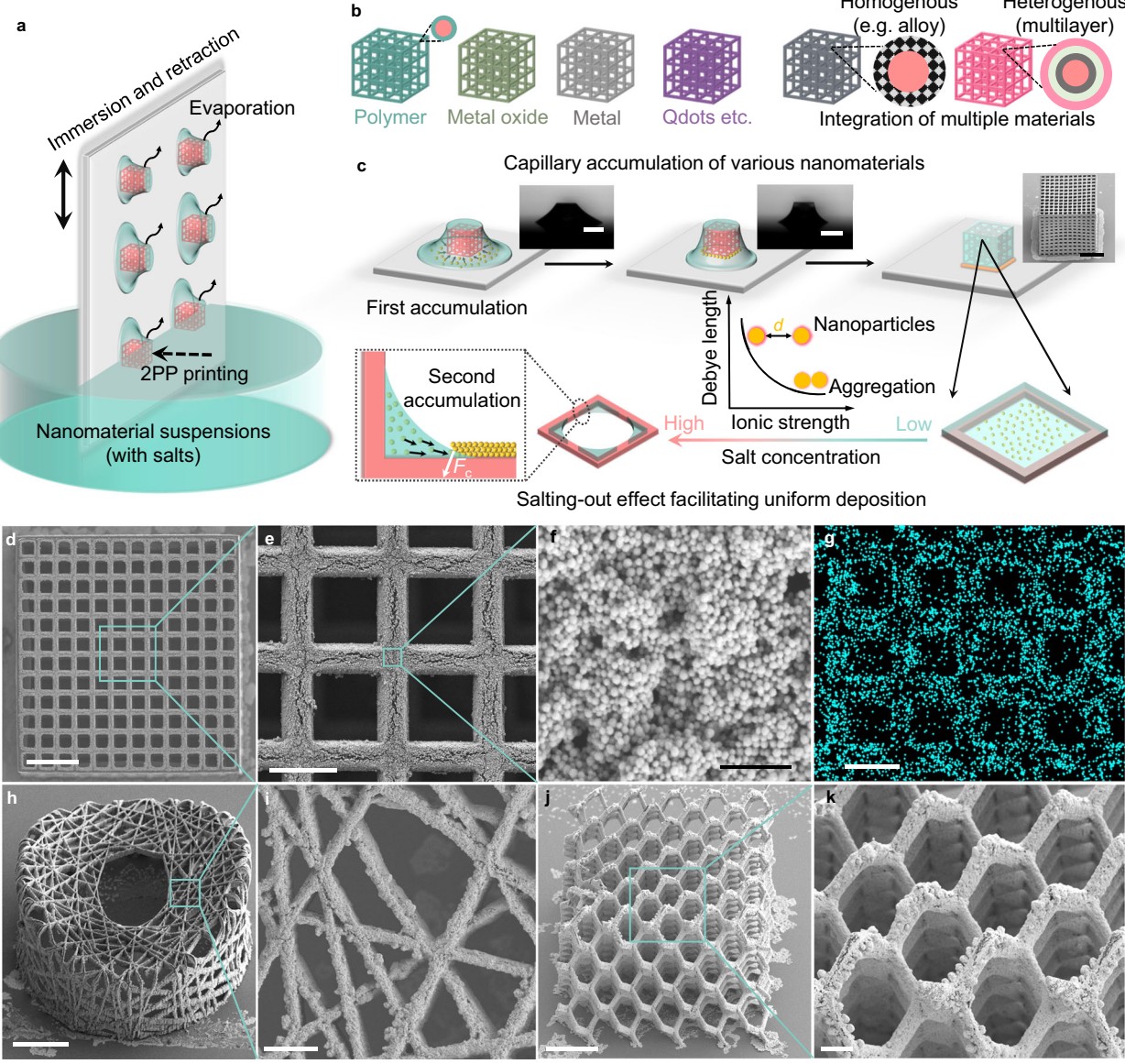

**Fig. 1 | Concept of capillary trapping-based 3D micro-/nanofabrication.**
**a**, **b** Schematic showing the capillary force-induced fabrication process of various nanomaterials. **c** Schematic illustration of key steps of capillary accumulation and deposition of nanomaterials: (I) the nanomaterials dispersed in bulk solution accumulate around the microscaffolds due to the strong pinning ability of the microstructure (first accumulation); (II) nanomaterials accumulate at the meniscus area on the beam of microscaffolds during solvent evaporation (second accumulation); (III) nanomaterials effectively coagulate and assemble onto the surface of the microscaffolds as the decline of electrostatic repulsion among nanomaterials.

Optical images (inset) show that solution is trapped by a microscaffold, and scanning electronic microscopy (SEM) image (inset) shows that Au NPs accumulate near a microscaffold. **d**–**g**, SEM images (**d**–**f**) and EDS mapping (**g**) of Au element distribution of a microscaffold assembled with Au NPs. Side-view SEM images (**h**, **i**) of a micro-architecture (bird's nest) and a 3D micro-metastructure (**j**, **k**) assembled with Au NPs. All samples underwent 600 immersion/retraction cycles. Scale bars are 300 μm for two optical images and 100 μm for SEM image in (**c**), 60 μm in (**d**, **h**, **j**), 20 μm in (**e**, **g**), 200 nm in (**f**), and 10 μm in (**i**, **k**).

resolution (Supplementary Fig. 5) and a roughness of 46.6 nm ± 3.9 nm (Supplementary Fig. 6), as well as can be widely applicable to assembling various non-specific nanomaterials, including polymeric and non-polymeric materials without needing to consider their specific physicochemical properties, such as their type, size, shape, surface charge, or surface functional groups. Furthermore, this approach is easily capable of integrating multiple materials either in a homogeneous or a heterogeneous manner, fabricating multifunctional 3D microstructures.

## Assembly dynamics and optimization of nanomaterial loading
To achieve a uniform, densely packed, and high-loading nanomaterial deposition on 3D microscaffolds, we investigate various factors

influencing the deposition quality during the first and second accumulation phases. Uniformly dispersed nanomaterials in the solvent is the prerequisite to ensure their uniform deposition onto the microscaffolds, and the most critical factor is the solution's high affinity to the 3D microscaffolds. For instance, isopropyl alcohol (IPA) demonstrates a high affinity for IPS polymer scaffolds, whereas $H_2O$ shows lower affinity. By varying the IPA/$H_2O$ ratio, we can finely adjust this affinity, as depicted in Supplementary Figure 3. With a low IPA concentration (<10% IPA/$H_2O$), where the solution has a contact angle greater than 55° with the IPS polymer, we observe that Au NPs tend to accumulate unevenly, especially at the corners of the microscaffolds' building blocks (Fig. 2a–c). This unevenness is attributed to the strong lateral force exerted by the liquid meniscus during the receding of the

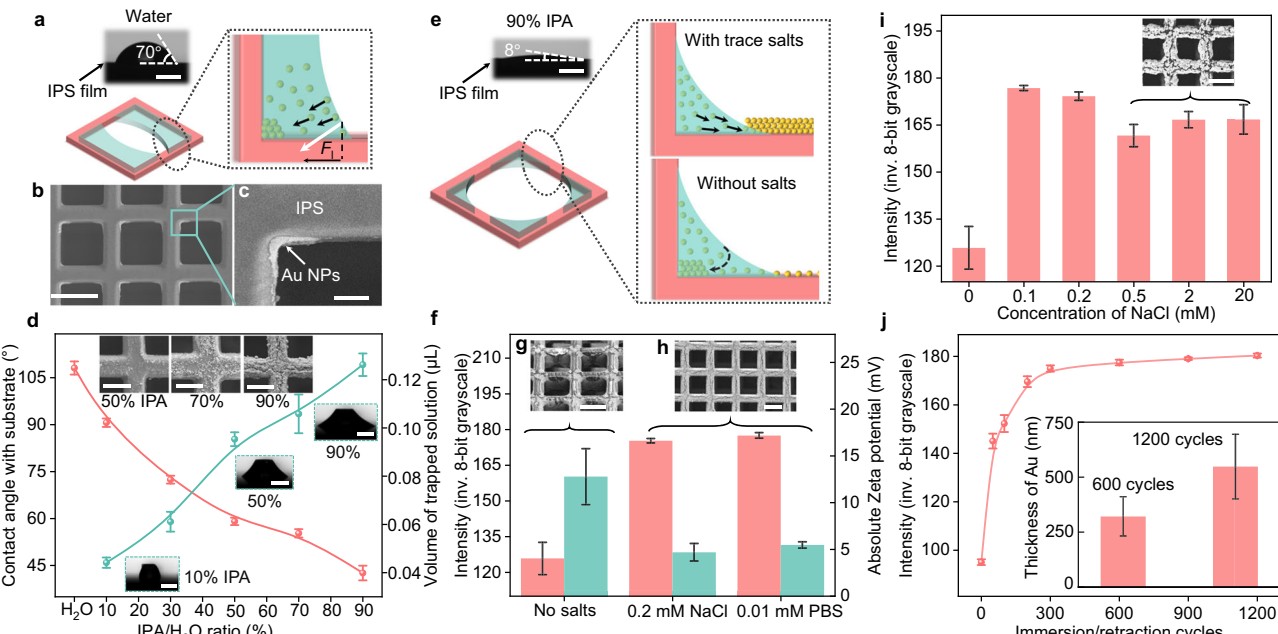

**Fig. 2 | Nanomaterial assembly dynamics and optimization. a** An optical image showing the water contact angle measurement on an IPS film and a schematic of the second accumulation phase. **b**, **c**, SEM images of microscaffold beams assembled with Au nanoparticles. **d** Contact angles (red dots) as a function of various IPA/H₂O ratio solutions on a silane-treated silicon substrate, alongside the trapped volumes of these solutions (green dots). Insets show optical images of solutions with different IPA/H₂O ratios trapped in a microscaffold and SEM images of Au nanoparticles assembled on the microscaffold beams through solutions with varying IPA/H₂O ratios. **e** Schematics of the second accumulation phase in a nanomaterial suspension with and without salts. **f** Intensity of averaged inverted 8-bit grayscale values (red bar) for microscaffolds treated with Au nanoparticles in 90% IPA/H₂O solutions containing different salts and their corresponding zeta potentials (green bar). SEM images of microscaffold beams assembled with Au nanoparticles in solutions without salts (**g**) and with 0.01 mM PBS (**h**). **i** Intensity of averaged inverted 8-bit grayscale values for microscaffolds treated with Au nanoparticles in 90% IPA/H₂O solutions with different NaCl concentrations. Inset shows an SEM image of microscaffold beams assembled with Au nanoparticles in a solution with 20 mM NaCl. **j** Intensity of averaged inverted 8-bit grayscale values for microscaffold beams assembled with Au nanoparticles over different immersion/retraction cycles. An inset depicts the thickness of the assembled Au NP layer after 600 and 1200 immersion/retraction cycles. All samples from (**b**) to (**i**) underwent 600 immersion/retraction cycles. Data in (**d, f, i, j**) are extracted from Supplementary Figs. 9, 12–15, respectively. These data points are shown as mean ± s.d. with at least 4 independent samples/measurements. ($n \geq 4$). Scale bars represent 300 μm for all optical micrographs in (**a, d, e**), 10 μm for SEM images in (**d**), 20 μm in (**b, g–i**), and 5 μm in (**c**). Source data are provided as a Source Data file.

contact line[48]. However, increasing the IPA/H₂O ratio above 50% substantially enhances the solution's affinity for the IPS polymer, leading to rapid spreading of the solution drops on the IPS polymer substrate (Supplementary Fig. 7). Consequently, the high affinity brings more uniform deposition of Au NPs as shown in the SEM images (insets of Fig. 2d). Additionally, the selective affinity of the silane-treated substrate and IPS polymer microscaffolds to the IPA/H₂O solution is also crucial. When the substrate is withdrawn from the solution, the amount of solution (and thus Au NPs) captured by the microscaffolds varies with the IPA/H₂O ratio. This variation results from the differentiated affinities of solution to the substrate. As the ratio increases, so does the solution volume captured by the IPS polymer microscaffolds (Fig. 2d), because of the reduced contact angle on the silane-treated substrate, leading to higher mass loadings of Au NPs. For optimal fabrication efficiency, we opt for a solution ratio of 90% IPA/H₂O in the subsequent experiments. In short, beyond the rational design of solvent used for the deposition (Supplementary Fig. 8), surface treatments of nanomaterials, microscaffolds, and substrates, including O₂ plasma treatment, chemical modification, or vapor deposition, can also enhance the efficiency and uniformity of the deposition, as these techniques can adjust their mutual selective affinity (wettability).

The surface charge of the nanomaterials is another crucial aspect for efficient nanomaterial deposition. Nanomaterials in solvents are typically charged, creating strong electrostatic repulsion as the solvent evaporates and the particles draw nearer to each other[49,50]. This repulsion impedes the effective coagulation and deposition of Au NPs, causing them to predominantly accumulate at the corners of the 3D

microscaffolds' building blocks as the contact line recedes, resulting in an uneven deposition. We find that adding a trace amount of salt can improve the nanoparticle deposition (Fig. 2e–h). Salt in the solution can effectively neutralize the surface charge of nanoparticles, reducing their electrostatic repulsion and facilitating their coagulation. For example, the presence of NaCl or phosphate buffered saline (PBS) in solutions results in a lower surface charge of Au NPs (zeta potential of ≈-5.0 mV) compared to solutions without salt (≈12.8 mV) (Fig. 2f). The salting-out effect, consequently, would aid in the dense packing and uniform deposition of nanoparticles. As illustrated in the SEM images of Fig. 2g, h, Au NPs show a more uniform and efficient deposition on 3D microscaffolds after evaporating the solution containing trace NaCl or PBS. We semi-quantify this effect by measuring the average (inverted) 8-bit grayscale of microscaffolds under an optical microscope. Different amounts of Au NPs cause distinct light absorption in the microscope, with scaffolds having less gold (before salt addition) appearing brighter (lower inverted grayscale value) and those with more gold (after salt addition) appearing darker (Fig. 2f, Supplementary Fig. 9). A small amount of salt increases the contrast in nanoparticle loading (Fig. 2i), but a higher salt concentration can adversely disrupt uniform deposition due to salt crystallization (Supplementary Fig. 10). Furthermore, any added salt can be easily removed by immersing the assembled microscaffolds in DI water for a few minutes (Supplementary Fig. 11).

Finally, the amount of the assembled nanomaterials can be gradually increased by repeating the immersion-retraction cycle. More cycles lead to darker microscaffolds (Fig. 2j), indicating increased Au

NPs loading. The average inverted grayscale value of the microscaffolds saturates after ≈300 immersion-retraction cycles, suggesting complete coverage of the microscaffolds beams by Au NPs. However, the thickness of the assembled Au NP layer keeps increasing after many cycles (inset in Fig. 2j). Given that the throughput of this approach is largely influenced by the number of immersion-retraction cycles, several strategies can be implemented to reduce the deposition time. First, increasing the concentration of nanoparticles in the solution can reduce the required number of cycles. Second, the drying time of the trapped solution during each cycle can be drastically shortened (from the current 50 s to less than 10 s) by implementing measures, such as reducing humidity, enhancing air ventilation, and raising the temperature with infrared light. Furthermore, several types of materials can be assembled automatically and simultaneously in our home-built platform, showing potential for scalable production (Supplementary Movie 2).

## High adaptability to versatile materials

Our capillary-trapping strategy for 3D micro-/nanofabrication stands out for its non-selectivity towards different nanomaterials. This bottom-up approach facilitates the incorporation of a diverse range of materials, as showcased in various 3D microstructures (Fig. 3). The 2PP printed IPS polymer 3D microscaffolds, initially smooth in morphology (Fig. 3a), acquire a rough texture upon dense nanoparticle assembly (Fig. 3b). When nanomaterials are modified with fluorescent molecules, their 3D structures can be visualized using confocal fluorescent microscopy. Our method allows for the assembly of various polymeric nanoparticles like polystyrene (PS, Fig. 3b) and poly (lactic-*co*-glycolic acid) (PLGA, Fig. 3c). Beyond polymers, our technique efficiently assembles non-polymeric nanomaterials, such as quantum dots (QDs, CdSe/ZnS; Fig. 3d), metals (Au nanorods, Ag NPs, Pt NPs; Fig. 3e–m), and metal oxides (Fe₃O₄ NPs, BaTiO₃ NPs; Fig. 3n–s). These materials are uniformly assembled onto 3D microscaffolds of various

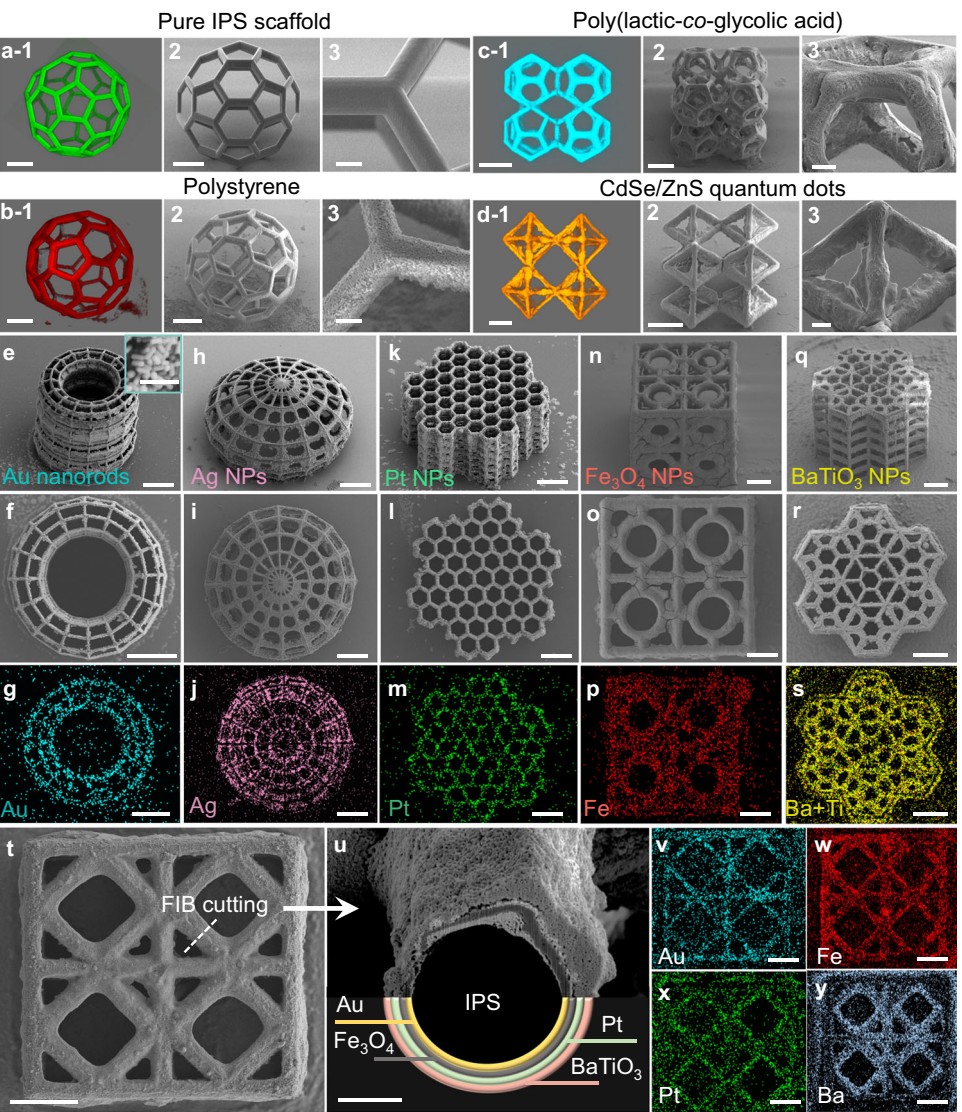

**Fig. 3 | High adaptability of the proposed fabrication process to versatile materials.** Fluorescent (1) and SEM images (2, 3) of the printed pure IPS (**a**) and PS nanoparticles-assembled (**b**) 3D hollow fullerene microstructures. Fluorescent (1) and SEM images (2, 3) of PLGA-assembled (**c**) and CdSe/ZnS quantum dots-assembled (**d**) diamond metastructures. **e–s**, SEM images and EDS mapping of various complex 3D microstructures assembled with diverse materials, including ring-shaped microscaffolds assembled with Au nanorods (**e–g**) and Ag NPs respectively (**h–j**), a honeycomb-like micro metastructure assembled with Pt NPs (**k–m**), a cube meta-structure assembled with Fe₃O₄ NPs (**n–p**), and a hexagon meta-structure assembled with BaTiO₃ NPs (**q–s**). **t** SEM image of a cube micro meta-structure after depositing four different materials. **u** Cross-section SEM image of a beam cut by focused ion beam (FIB) from the structure in (**t**). The order of assembled material from inner to outer is Au, Fe₃O₄, Pt, BaTiO₃. **v–y**, EDS mapping of Au, Fe, Pt, and Ba elements. The experimental parameters of such various nanomaterials are summarized in Supplementary Table 3. Scale bars are 10 μm in (3 of **a–d**), 200 nm in (inset in **e**), 5 μm in (**u**), 60 μm in all others.

geometries, confirmed by their characteristic EDS mappings. This adaptability extends to nanomaterials of varying sizes (from 5 to 500 nm), shapes (e.g., spheres, rods), and surface groups (e.g., -COOH, -NH$_2$, PEG, alkyne, azide), enabling on-demand functionality (Supplementary Table 2).

Moreover, our approach can fabricate 3D microstructures with a uniform distribution of multiple materials, which surpasses the existing single-material strategies[36,37]. This flexibility allows for the creation of heterogeneous, multi-layer, and multifunctional material structures. We achieve this by simply replacing the suspensions with different nanomaterials during sequential immersion/retraction cycles. To showcase the potential for integrating multiple materials into a single 3D architecture, we deposit four different nanomaterials (Au, Fe$_3$O$_4$, Pt, BaTiO$_3$) sequentially onto a single 3D microstructure (Fig. 3t–y). The cross-section SEM image shown in Fig. 3u, obtained from a focused ion beam (FIB)-cut beam, exhibits four distinct contrasts, each indicating a separate material layer. Further, the EDS mapping confirms that elements, such as Au, Fe, Pt, and Ba, are uniformly distributed within the microscaffolds (Fig. 3v–y).

## Mechanical properties of the as-fabricated structures

The structural stability of our 3D microstructures can be enhanced through thermal annealing (Fig. 4, Supplementary Movies 3, 4), which shows the mechanical performance of nanomaterial-assembled microstructures before and after thermal annealing at different temperatures during compression tests. Initially, these structures are primarily stabilized by van der Waals forces, which are less robust than covalent or ionic bonds[29]. To enhance their interaction and strength, we implement a post-annealing process. Given that nanomaterials typically have lower melting points than their bulk counterparts, we can achieve stronger interfacial bonds through low-temperature sintering[29,37,51]. This process is feasible because the IPS polymeric scaffolds we employ are thermally stable below 300 °C. The interfacial melting of the nanoparticles during annealing remarkably enhances the mechanical strength of the structures. For example, an IPS microcube assembled with Au NPs was annealed at 120 and 240 °C, causing the Au NPs to partially melt and reconfigure into a dense,

continuous 3D porous Au upon cooling (Fig. 4a). Both samples annealed at these temperatures (with a density of ≈4.18 g cm$^{-3}$) exhibited a higher compressive strength than their untreated counterpart (Fig. 4b) with enhancements of 160% and 120% in their maximum compressive strength for the respective temperatures. However, it is important to note that higher sintering temperatures do not invariably result in increased mechanical strength; excessive melting of Au nanoparticles can lead to uneven nanoparticle coagulation or potential degradation of the IPS scaffold.

When the annealing temperature exceeds the degradation temperature of polymer scaffolds, the polymer can be completely removed, resulting in a 3D microstructure composed solely of the targeted materials. For instance, Fig. 4e show the morphological comparison of a 3D fullerene-like microsphere assembled with Fe$_3$O$_4$ NPs before and after annealing at 600 °C for 2 h. The annealed structure maintains good shape integrity with slight shrinkage and deformation, despite the removal of the polymer scaffold. The key to preserving the high integrity of these annealed microstructures at elevated temperatures is ensuring a uniform and high loading of nanoparticles on the microscaffolds. This uniformity is crucial as non-uniform or sparse distribution of nanomaterials can lead to structural instability, especially due to the thermal degradation of the IPS polymer scaffold. A typical example of this issue is observed in the same type of 3D fullerene-like microsphere, which collapses after annealing at 600 °C due to a lesser load of Fe$_3$O$_4$ nanoparticles (Supplementary Fig. 16). Once the polymer template is removed, the pure materials exhibit increased rigidity; this is evidenced by the steeper force profiles observed at higher annealing temperatures (Fig. 4f).

Heterogeneous 3D microstructures can be fabricated by co-deposition multiple nanomaterials. Figure 5a–g show the co-deposition of Au and Ag on a hexagon microstructure. After annealing at 240 °C for 2 h, a co-melting morphology revealed by their EDS mapping (Fig. 5e, f) demonstrates the potential for creating alloys-assembled microstructures. In addition, this technique is allowing for the creation of the micro-/nanodevices with diverse materials, equipped with on-demand functionalities. An example is the multifunctional 3D fullerene-like microrobots made entirely of a Fe$_3$O$_4$-BaTiO$_3$

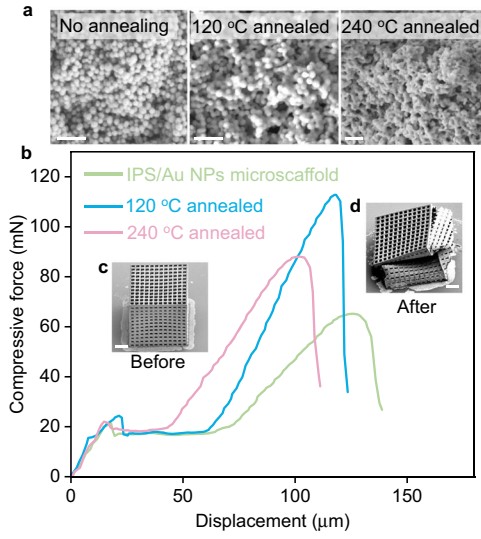

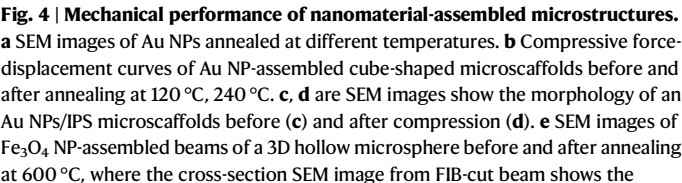

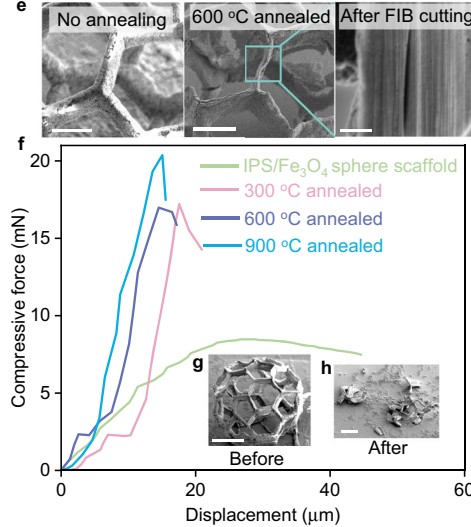

**Fig. 4 | Mechanical performance of nanomaterial-assembled microstructures.**
**a** SEM images of Au NPs annealed at different temperatures. **b** Compressive force-displacement curves of Au NP-assembled cube-shaped microscaffolds before and after annealing at 120 °C, 240 °C. **c, d** are SEM images show the morphology of an Au NPs/IPS microscaffolds before (**c**) and after compression (**d**). **e** SEM images of Fe$_3$O$_4$ NP-assembled beams of a 3D hollow microsphere before and after annealing at 600 °C, where the cross-section SEM image from FIB-cut beam shows the

polymer scaffold is removed. **f** Compressive force-displacement curves of Fe$_3$O$_4$ NP-assembled microspheres before and after annealing at 300, 600, and 900 °C. **g, h** are SEM images showing the morphology of a 900 °C annealed Fe$_3$O$_4$ NPs microsphere before (**g**) and after compression (**h**). Scale bars are 200 nm in (**a**), 60 μm in (**c, d, g, h**), 30 μm in (**e**, SEM images of "no annealing" and "600 °C annealing"), 1 μm in (**e**, cross-section SEM image). Source data are provided as a Source Data file.

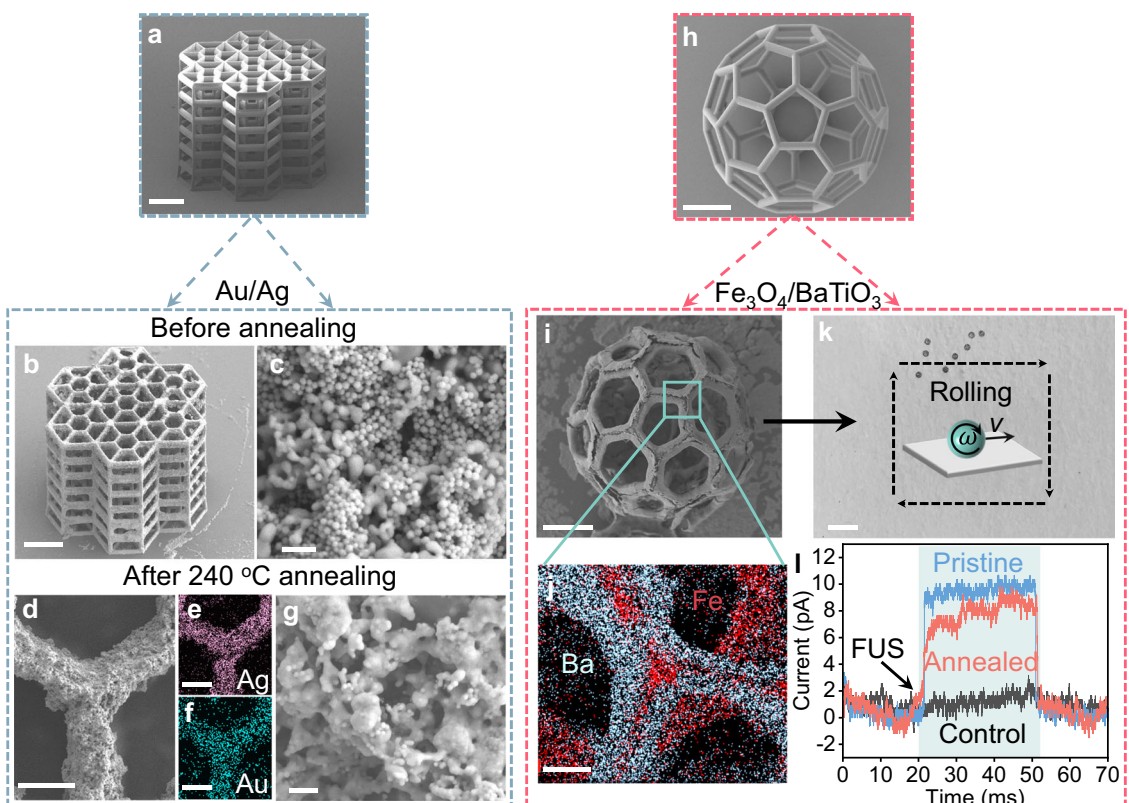

**Fig. 5 | Fabrication of microstructures made of heterogeneous materials and demonstration of on-demand functionalities.** SEM images of an IPS honeycomb-like microscaffolds (**a**), the Au NPs-Ag NPs co-assembled (**b**), where the morphologies of Au, Ag NPs on the beams before annealing (**c**) and after annealing treatment (240 °C for 2 h, **d**, **g**), as well as the EDS mappings of Ag (**e**) and Au (**f**) elements distribution on beams after annealing treatment. SEM images of IPS (**h**) and $Fe_3O_4$/$BaTiO_3$ co-assembled (annealed at high temperature of 600 °C, (**i**)) 3D hollow microsphere. **j** EDS mapping of Ba and Fe elements distribution. **k** Video snapshot showing magnetically controlled microrobots made of $Fe_3O_4$-$BaTiO_3$ 3D hollow microspheres (annealed at 600 °C). **l** Piezoelectric current of the microrobots under the treatment of a focused ultrasound with a power of 100 mW m$^{-2}$. Scale bars are 60 μm in (**a**, **b**, **h**, **i**), 200 nm in (**c**, **g**), 10 μm in (**d**–**f**, **j**), and 1 mm in (**k**). Source data are provided as a Source Data file.

mixture, which retains their structural integrity with magnetic and piezoelectric properties (see Methods for the characterization details) after annealing at 600 °C to remove the polymer scaffolds (Fig. 5h–j). These annealed $Fe_3O_4$-$BaTiO_3$ microrobots can be magnetically driven (contributed from $Fe_3O_4$) and manipulated as demonstrated in Fig. 5k and Supplementary Movie 5, and exhibit piezoelectric property (contributed from $BaTiO_3$) as well. The annealing process does not noticeably degrade the piezoelectric performance as the piezoelectric current is ≈8.0 pA (originally ≈ 9.5 pA) under the treatment of focused ultrasound (FUS) with a power of 100 mW m$^{-2}$ (Fig. 5l). These multifunctional microrobots can be wirelessly manipulated and used for targeted electrical stimulation, demonstrating great potential for biomedical applications, particularly in targeted cell stimulation[52]. Our approach to 3D micro-/nanofabrication of various materials introduces collective or synergistic functionalities, which holds promise for tailored applications, including the development of advanced smart microrobots and microdevices.

## Discussion

To summarize, our study introduces a versatile 3D micro-/nanofabrication method that can construct diverse 3D micro-/nanostructures using various nanomaterials. This technique harnesses the capillary force—a widespread, non-specific physical interaction on solid-liquid-gas three-phase interfaces—to gather nanomaterials and guide their assembly onto intricate 3D microscaffolds. Unlike conventional assembly methods that rely on other physical interactions, such as electrostatic forces[36] and size-dependent steric effects[37], our capillary

force-driven approach offers broader applicability to both polymeric and non-polymeric materials. It also ensures uniform deposition and high material loadings. Crucially, this method allows the integration of multiple functional materials into a single 3D microstructure, especially in a layer-by-layer fashion. By appropriately choosing multimaterials and designing their deposition order, we envision that this capability is able to create advanced micro-/nanodevices with customizable, on-demand multifunctionalities, for example, single stimulus-powered multifunctionalities on a microrobot[53], or a single multifunctional microrobot that can be powered by versatile stimuli (ultrasound, light, electromagnetic field, and chemical reactions).

Although we have demonstrated the on-demand fabrication of versatile materials into 3D micro-/nanostructures, several challenges remain for our future research. For instance, it has been observed that these micro-/nanostructures are often composed of a relatively rough surface and random assembly of nanoparticles, which could potentially result in inferior properties compared to their bulk counterparts. Additionally, limited to the mechanism of the capillary capture of nanomaterials, our strategy primarily facilitates the creation of 3D scaffolds rather than achieving true volumetric 3D structures (see Supplementary Fig. 2). Furthermore, the exploration of milder etching technologies, such as plasma and wet etching, is necessary to extend the material applicability for fabricating pure-material microstructures (as some material would lose their functionality at high temperatures, e.g., QDs[54,55]), improve the morphology and structural integrity of the pure-material micro-/nanostructures. Overall, our strategy in creating 3D micro-/nanostructures across a wide spectrum of materials leads to

a variety of tailored functionalities, stemming from the dynamic interplay and flexible synergy in both intricate structural and material designs, which holds promise for advancing the fields of micro-robotics, microelectronics, and micro-/nanooptics.

# Methods

## Materials

Gold nanoparticles (Au NPs) (dispersed in 0.1 mM PBS, reactant free, 5-400 nm), Au NPs (alkyne-functionalized, 50 nm), Au NPs (azide-functionalized, 70 nm), Au nanorods (25 nm × 60 nm, CTAB as stabilizer, dispersed in water), CdSe/ZnS core-shell quantum dots (carboxylic acid functionalized, dispersed in water, fluorescence $\lambda_{em}$: 620 nm), iron oxide ($Fe_3O_4$, amine, carboxylic acid, and PEG functionalized, dispersed in water, 30 nm, 1 mg mL$^{-1}$,), silver (Ag) powder (<100 nm, PVP as dispersant, 99.5%), copper (Cu) nanopowder (25 nm), platinum (Pt) nanopowder (<50 nm), orange fluorescent PLGA nanospheres (excitation/emission: 530/582 nm, 100 nm), barium titanate ($BaTiO_3$) nanopowder (50 nm, cubic), titanium (IV) oxide ($TiO_2$, anatase, <25 nm), diamond nanopowder (97%, <10 nm), trichloro(1H,1H,2H,2H-perfluorooctyl) silane (97%), and sodium chloride (NaCl, 99%) were purchased from Sigma Aldrich. Isopropyl alcohol (IPA, 99.9%) was purchased from Carl Roth GmbH. IPS photoresist was purchased from Nanoscribe GmbH. Red fluorescent polystyrene (PS) nanospheres (500 nm) was purchased from Microparticles GmbH. $BaTiO_3$ (99.95%, 280 nm, tetragonal) was purchased from Blografi. Polyacrylic acid (PAA)-coated upconverting nanoparticles (30-35 nm) were purchased from CD Bioparticles. Phosphate buffered saline (PBS, pH 7.4, 1×) was purchased from Thermo Fisher Scientific.

## Fabrication of 3D microscaffolds

All 3D microstructures in this work are printed using a commercial direct laser writing system (Photonic Professional GT, Nanoscribe GmbH) with a 25× objective and IPS photoresist (a methacrylate-based resin). The laser power and printing speed are 50 mW and 50,000 μm s$^{-1}$ respectively. These microscaffolds are printed on a silane-treated silicon substrate (treated by trichloro(1H,1H,2H,2H-perfluorooctyl) silane vapor for overnight). After the printing process, the substrate was developed in IPA solution for 5 min to remove non-polymerized photoresist. These 3D micromodels are designed by Solidworks 2021 or Cinema 4D.

## Nanomaterial deposition

The deposition experiments were performed at room temperature with a homebuilt immersion/retraction system (Supplementary Movie 2), which is operated automatically with the Python codes. The initial nanomaterial suspension is first centrifuged at ≈10,000 × g, followed by discarding the supernatant to decrease the concentration of surfactants from the original solution. Then a solution with a certain concentration of nanomaterials (e.g., 1.2 mg mL$^{-1}$ Au NPs) comprising a 90% volume ratio of IPA (denoted as 90% IPA/$H_2O$) and 10% 0.1 mM PBS is prepared. In an immersion/retraction cycle, the substrate with printed microstructures is immersing into the nanomaterial dispersions, followed by a retracting process with a speed of ≈3 mm s$^{-1}$ until the substrate is entirely outside of the container. After waiting for 50 s for a fully evaporation of the trapped solution, the substrate then re-immerses into the suspension to resume the following immersion/retraction cycle. According to the concentration of nanomaterials, size, and complexity of microstructures, the waiting time and immersion/retraction cycles can be rationally tuned by modifying the codes to improve the deposition efficiency. Furthermore, the system can deposit multiple materials onto several substrates simultaneously (Supplementary Movie 2), showing potential for scalable production. The concentration of various nanomaterials used in the main text and their immersion/retraction cycles are summarized in Supplementary Table 3.

## Sintering and annealing post-treatment

To obtain more robust microstructures, the microstructures after nanomaterial deposition are sintered at 240 °C for 2 h in the air at a heating rate of 2 °C min$^{-1}$. When further increasing the annealing temperature above 600 °C, the IPS polymer scaffold can be completely removed.

## Microscopy characterization

All fluorescent images were captured by a SP8, Leica confocal microscope, and the raw data were processed and colorized via the built-in software LAX. The excitation wavelength in all experiment is set to be 552 nm with 2% intensity, and emission wavelength is 640–700 nm. With these parameters, no fluorescent signals from the IPS template are detected. Therefore, the targeted fluorescent signals from PLGA, red PS, and CdSe/ZnS core-shell quantum dots can be distinguished and recorded. SEM and EDS images were obtained using a Gemini field-emission SEM 500 at an acceleration voltage of 5 or 20 kV. FIB-SEM cross-section results were obtained via the FEI Nova 600 Nano Lab FIB-SEM system with a current value between 0.1 and 3.0 nA. An atomic force microscope (NanoWizard 4, JPK Instruments) was used to measure the roughness of the fabricated microstructure.

## Characterization of mechanical properties

A customized mechanical characterization setup was used for compression measurements. The video camera (Grasshopper3, Point Gray Research Inc.) was connected to an inverted optical microscope (Axio Observer A1, Zeiss). A computer-controlled high-precision stage (LPS-65 2″, Physik Instrumente GmbH & Co. KG) was attached to the microscope in the z-direction. For measurements of forces on the z-axis, a load cell (GSO-25, Transducer Technique LLC) was mounted on the setup. The motion of the piezo stages was controlled, and a custom-made program was used for data acquisition processing on a LabVIEW (National Instruments, Austin, TX, USA). In the x-axis and y-axis, a manual xy-axis stage (NFP-2462CC, Positionierungstechnik Dr. Meierling) was used for fine positioning, and two goniometers (M-GON65-U, Newport) were utilized for adjustment of tilt correction. A smooth hemispherical glass probe with a 10 mm diameter was used as a contact surface. Since the size of the hemispherical probe is much larger than the microstructure samples, local flat contact can be assured during the measurements. Then the sample is attached to the setup through a flat-end screw. During the measurements, the attached sample was lowered slowly onto the smooth hemispherical probe; after the test, it moved upwards at a controlled speed. The approaching and retraction speeds were set to 50 μm s$^{-1}$ on the z-axis. These measurements were conducted in controlled conditions (23 °C, 30% rel. humidity).

## Zeta potential measurement

Au NPs were dispersed in IPA/$H_2O$ solution containing different concentration of IPA and salt, and their Zeta potentials are measured in a system of WYATT Mobius. The viscosity and relative permittivity of solutions with various volume fractions of IPA were set according to ref. 42. for the measurement. The Zeta potential of Au NPs dispersed in different solutions is obtained by averaging the results of 5 independent measurements.

## Contact angle measurement

The contact angle measurements were conducted at room temperature using a commercial device (Drop Shape Analyzer DSA100, Krüss GmbH). Specifically, IPS films were first prepared according to approaches reported in ref. 56, where the IPS photoresist was injected into a glass capillary cell of ≈200 μm height and then polymerized under 365 nm UV light for 1 h. 2 μL 0.01 mM PBS solutions with different volume fractions of IPA (without nanomaterials in solution) were dropped on a silane-treated silicon substrate and an IPS film

respectively. The optical image from side perspective were captured and analyzed by the built-in software ADVANCE. The final contact angle is obtained by averaging the results of 5 independent measurements.

## Semi-quantification of loading amount

After depositing varying amounts of Au NPs, the microscaffolds exhibited distinct light absorption patterns when observed under an optical microscope, as demonstrated in Supplementary Figs. 9, 14. To semi-quantitatively assess the differences in light absorption among these samples, optical micrographs of micro-frameworks with different amounts of Au NPs were initially captured using a Zeiss Axio Imager 2 microscope. All imaging parameters, including light intensity and exposure time, were consistently maintained. These micrographs were then converted into 8-bit grayscale images using the Fiji software, facilitating the extraction of the average grayscale for each beam. For this analysis, five beams situated at identical positions on different microscaffolds were selected to calculate their average grayscale. Subsequently, the average inverted grayscale values were determined by subtracting the average 8-bit grayscale values from 255. Consequently, micro-frameworks with a higher concentration of Au NPs, which appear darker under the microscope, correspond to higher average inverted grayscale values.

## Estimation of the trapped solution volumes by a microscaffolds

Given that the shape of the solution trapped by a microscaffolds closely resembles a frustum of a cone, the volume of the trapped solution can be estimated to Eq. (1):

$$V = (\pi h / 3)(R^2 + Rr + r^2) \tag{1}$$

where $r$ and $R$ represent the radii of the smaller and larger bases of the frustum, respectively, and $h$ is its height. In this context, considering the dimensions of the microscaffolds (a cube measuring $282\,\mu m \times 282\,\mu m \times 282\,\mu m$), $h$ is $282\,\mu m$. The radius $r$ is approximately half the length of the diagonal of the upper square base, which is around $200\,\mu m$. The constant $\pi$ is approximated as 3.14. The value of $R$ changes with the increasing IPA/$H_2O$ ratio. It can be determined by measuring the area of the circular lower base using Fiji software, as illustrated in Supplementary Fig. 7.

## Estimation of the thickness of the assembled Au NP layer

The cross-section SEM image of a Au NP-assembled beam was first acquired after the FIB treatment. As shown in Supplementary Fig. 15, a clear interfacial contract between the assembled Au NP layer and the inner polymer scaffold was observed due to the huge difference in the conductivity of Au and polymer beam. The thickness of the assembled Au NP layer is determined by averaging thickness data extracted from different positions of the beam using the software of Fiji.

## Estimation of the density of the fabricated structures

To obtain the density of nanomaterial-assembled microstructures, a cube-shaped microscaffolds assembled with Au NPs after 600 immersion/retraction cycles is taken as an example. The thickness of the assembled Au NP layer is $\approx 0.32\,\mu m$. The density of IPS polymer ($\rho_{IPS}$) is $1.19\,g\,cm^{-3}$, and the density of Au ($\rho_{Au}$) is $19.32\,g\,cm^{-3}$. The volumes of a pure IPS cube-shaped microscaffold ($V_{IPS}$), an Au NP-assembled microscaffolds ($V_t$), and the assembled Au NP layer ($V_{Au}$) can be calculated from the cube model and measurement, and the density of the assembled cube can thus be estimated as:

$$\rho = \frac{\rho_{IPS} \cdot V_{IPS} + \rho_{Au} \cdot V_{Au}}{V_t} \approx 4.18\,g\,cm^{-3} \tag{2}$$

## Magnetic manipulation of the microrobots

A permanent NdFeB cube magnet, measuring $12\,mm \times 12\,mm$, is affixed to a rotating motor to facilitate the actuation of magnetic fullerene-like microrobots. The magnetic field strength generated by this setup is $\approx 35\,mT$.

## Piezoelectric current measurement

The piezoelectric current generation of $Fe_3O_4$-$BaTiO_3$ microrobots was measured in an interconnect-free setup using a patch-clamp system under low-intensity focused ultrasound (LIFU)[57]. An in-house measurement system was employed for this purpose. This system includes a patch-clamp amplifier (Axopatch 200B, Molecular Devices, CA, USA) connected to a low-noise data acquisition system (Axon Digidata 1550B, Molecular Devices, CA, USA). Measurement glass pipettes pulled using a P-2000 micropipette puller (Sutter Instrument, CA, USA) to achieve a resistance of 10-12 MΩ, were utilized. The measurement chamber was filled with a fresh ionic medium (Invitrogen™ Live Cell Imaging Solution (LCIS), Fisher Scientific), into which the assembled structures were added. These measurement pipettes were filled with an intracellular solution (Internal KF 110, Nanion, Munich, Germany) for operation in voltage-clamp mode. An Olympus microscope equipped with a Prime BSI Scientific CMOS (sCMOS) digital camera monitored the procedure. The Ag/AgCl ground electrode, immersed in the ionic medium, served as the measurement ground. The LIFU transducer, situated in an acoustic tank filled with deionized water, was directed toward the center of the measurement chamber. The glass measurement pipettes were positioned near the printed structures (as shown in Supplementary Fig. 17), and the system voltage was held at zero in voltage-clamp mode to create a virtual ground, in accordance with the published procedure. A focused ultrasound intensity of $100\,mW\,cm^{-2}$ was applied to the recording chamber. Data was processed using a $1\,kHz$ low-pass filter and sampled at a rate of $50\,kHz$.

## Reporting summary

Further information on research design is available in the Nature Portfolio Reporting Summary linked to this article.

# Data availability

The data that support the findings of this study are available from the corresponding authors upon request. Source data are provided with this paper.

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

## Acknowledgements
The authors thank Dr. Jie Han and Saadet Baltaci for their technical assistance. The authors thank Ulrike Eigenthaler for her help in FIB characterization. This work was funded by the Max Planck Society, Chinese Scholarship Council (X.L.), Alexander von Humboldt Foundation (M.Z.), and European Research Council (ERC) Advanced Grant SoMMoR project with grant no. 834531 (M.S.).

## Author contributions
M.Z. conceived the idea and proposed the research; X.L., M.Z., M.S. designed and planned the research; X.L. performed the experiments with the technical assistance from Z.Z., A.S., M.H., C.B.D., M.Z.; Z.Z. contributed to the design of 3D models and the assembly of the experimental setup. X.L. performed data analysis; M.Z., M.S. supervised the research; X.L., M.Z. wrote the original draft with the edits from all authors.

## Funding

## Competing interests
The authors declare no competing interests.
