## [Peer Review File · Nature Communications]

Capillary trapping of various nanomaterials on additively manufactured scaffolds for 3D micro-/nanofabricationREVIEWER COMMENTS

Reviewer #1 (Remarks to the Author):

The manuscript describes a strategy to address the materials limitation in 3D printing, harnessing TPP for 3D printing and multiple “immersing and retracting” process to coat nanoparticle layers on the 3D skeletons. The authors demonstrated the manufacturing of diverse materials, heterostructures, and the mixture. They also investigated the thermal treatment on the sintering of nanoparticles and removal of polymers. They provided an application example on 3D microrobot using magnetic nanoparticles for demonstration.

The 3D printing of inorganic materials is important for various device applications. However, using 3D-printed polymers as skeletons for further deposition/adsorption has been reported in the past one/two decades, see a recent paper in Nature 612, 685–690 (2022). Using “capillary trapping” is definitely different from previous strategies such as PVD, chemical coating, electrochemical coating, or adsorption, while the advantage and drawback should be clearly elaborated in the paper. There are also some papers demonstrating direct laser 3D printing of colloidal nanocrystals, such as Science 377, 1112-1116 (2022); Science 381, 1468-1474 (2023) (refs 21 and 30 cited in the paper). While the authors said that the printing throughput is limited in the refs, how about other key parameters such as printing resolution, mechanical properties, ..., compared with the developed techniques. Moreover, I also spot that the discussion on ref. 21 is inaccurate in the manuscript, as no photoresist or monomer was used in that paper. Overall, the careful discussion on the printing capability of both the developed technique and other approaches should be done.

I have some other comments on the papers, which should be well addressed before the paper can be accepted for publication.

1. The interparticle van der Waals force is basically a weak force. What is the mechanical properties of the fabricated 3D structures? Without sintering, how the weak van der Waals force overcomes the capillary force during evaporation?
2. What is the resolution of this approach? What is the minimum thickness of the coated nanoparticle layer? What is the density of the fabricated structures? What is the surface roughness? These key factors are important for the future application of the technique.
3. The coating of nanoparticles on the skeletons relies on the wettability. That means the surface chemistry of both the nanoparticles and the 2PP-printed 3D skeletons, as well as the solvent, should be important. I expect more discussion here.
4. The versatility of the proposed techniques should be carefully backup. For instance, how to make a volumetric 3D structures with high purity? It should be clearly pointed out what can be done and what cannot be achieved, as we know no technique is perfect.
5. I can spot some defects generated in Fig. 4 after annealing at 600-700 oC. However, I did not see discussion on the structural change and the impact on mechanical properties and applications.

Reviewer #2 (Remarks to the Author):

The manuscript titled "Capillary trapping of wide-spectrum nanomaterials for three-dimensional micro/nanofabrication" by Lyu et al. describes a processing route for multi-material deposition of colloidal systems on printed 2PP frameworks/scaffolds.

The work is based on the vision that the proposed processing route offers a new technology for forming multi-material components in a size range that cannot be achieved by methods other than 2PP. To support this opinion, it is stated that 2PP is not capable of processing materials other than polymers. In this context, it states (line 54):

“On the other hand, the materials printed through these chemical or physical strategies are either chemically bonded with organic photoresist initiators or embedded in a polymer matrix, which can limit the functionality of the resulting microstructures. Developing a highly efficient 3D micro/nanofabrication strategy that is compatible with any non-specific materials, such as polymeric and non-polymeric materials, remains a significant challenge.”

In fact, this statement is not true. ref 28 essentially follows a conventional ceramic processing route and delivers 3D microstructures of high-performance ceramics after sintering. This process and the products are no different from continuously produced ceramics, in fact they are better as the structural features are smaller than the critical defect size. In this sense, I do not see the argument here that the 2PP molded ceramics are inferior to standard high performance ceramics. Furthermore, polymer-derived ceramics and sol-gel technologies in combination with 2PP enable the production of advanced functional materials, as has been proven by a considerable body of literature devoted to this topic.

Furthermore, I do not see that the impregnation of frameworks/scaffolds is new. Impregnation of foams is a widely used technology that is very similar to the methodology presented, except that the foams are not 2PP molded. After firing/sintering the foams disappear. On this particular point, I admit that the work represents a new approach, as it uses 2PP for the manufacture of scaffolds, however, the results obtained are not convincing and are at a preliminary stage. Unlike existing technologies, which generally follow the same strategy, the parts presented in this study do not have advanced characteristics.

Reviewer #3 (Remarks to the Author):

The work described in the manuscript addresses an original approach for nanoparticle assembly using capillary forces in a 3D printed micro scaffold. Overall this is great work and truly new, and certainly forward looking with new opportunities for nanomaterial assembly into functional constructs. The authors have asked the key scientific questions and addressed many issues that can be expected, but well addressed, or unforeseen and then smartly studied for their influence on the and also addressed.

The strength of the paper is its quite simple but strong and generic concept idea, and the execution of a battery of parameter screening. The authors have used the now well-established 2PP for scaffold printing, and then assembled a multitude of mono- or up to 4 heterogenous material in a sequential manner. Process was optimized to increase assembly control and yield by solvent modification for contact angle tuning, and fishing/drying timing was also studied systematically for optimal layer growth, as well as adding salt and post assembly annealing. The grown film were then analyzed with fluorescent, TEM and other microscopic techniques to quantify, sometime semi-quantitatively, the quality and metrology of the NP assembly. First order validation of fabricated devices include the motion under an electric field of magnetic nanorobots made with the 2PP-assembly method.

The text reads well, and the figures are informative and concise.

The idea of the work clearly qualifies for a paper in this journal, but would need a few fixes and additions that seem to be missing, and that would complete the study for a larger readership.

/The work is based on 2PP made scaffolds. In most cases the printed scaffold remains under the assembly, or is removed by the high temperature sintering. This means that the costly scaffold can only be used once. Is there no way to preserve the scaffold for multiple assemblies, so that this opens a route for mass manufacturing of the built nanosystems?

/some immersion-fishing cycles go up to several hundreds. What is the time this needs to be done. Maybe the time scale could be more clearly highlighted (

/it is not entirely clear what the purpose of the BaTiO₃ buckyball is. The piezoresponse results are not well described and leaves the reader a bit alone.

/the melting temperature of Au NP at 50 or 70nm is certainly higher than the used 240 degC, so this reviewer is not sure that this will lead to a melting of the particles.

/NP colloidal solutions need surfactants to prevent agglomeration. What happens with the surfactants after the assembly? Could it still be there and 'pollute' the material?

/what is the authors observation of assembled NP from a previous step are eventually repositioned or removed in a subsequent step, knowing that capillary forces are strong.

/if NP are uniform in size distribution, they can form crystalline features. Any comment or observation? How does the 2PP 3D printed scaffold influences crystal like assembly?

/is there a way to speed up the drying step in between the dipping steps, e.g. by some gentle heating or would it compromise the dynamics of the drying process for the assembly?

Reviewer #4 (Remarks to the Author):

This manuscript demonstrates a versatile and easy to adapt process to coat 2PP-printed micro-and nanostructures with a variety of nanoparticles. The materials span from metal over low sintering temperature ceramics and polymers to quantum dots. The adjustment of the suspension for the scaffold wetting and the surface charges of the nanoparticles allows for a uniform coagulation of the particles at the structures during the drying process.

The concept shows a simple way to produce even functional devices but comes with a few flaws and drawbacks, which must be discussed carefully.

1. The process is a form of dip-coating on structures materials with particle assembly during drying. Why is the word "coating" first and only used in line 194?

2. The variety of materials, which can be coated, is impressive, but the authors should be careful. The resulting structures do not possess the same properties as if they would be completely consisting of either metal, ceramic or a polymer, as those are core-shell-structures.

3. The authors should also discuss their results compared to differently coated and hollow 2PP-structures like:

a. Formanek, F., Takeyasu, N., Tanaka, T., Chiyoda, K., Ishikawa, A., & Kawata, S. (2006). Three-dimensional fabrication of metallic nanostructures over large areas by two-photon polymerization. *Optics express*, 14(2), 800-809.

b. Bauer, J., Hengsbach, S., Tesari, I., Schwaiger, R., & Kraft, O. (2014). High-strength cellular ceramic composites with 3D microarchitecture. *Proceedings of the National Academy of Sciences*, 111(7), 2453-2458.

c. Meza, L. R., & Greer, J. R. (2014). Mechanical characterization of hollow ceramic nanolattices. *Journal of materials science*, 49, 2496-2508.

4. The 2PP-printed scaffolds are from a commercial resin with unknown ingredients. Most likely they are silicon-based pre-ceramic polymers. During annealing the organic part indeed burns off but the glassy silicon oxide remains. There is no proof shown, that the used polymer/preceramic polymer is removed (EDS/EDX for Si?).

5. Line 12-13: The introduction sentence "Fabrication of three-dimensional (3D) micro/nanostructures with wide-spectrum materials has been increasingly important across various fields." comes in a little bumpy. The fabrication is still important, not was and a "wide range of materials" and "across various fields" are rather generic phrases. Why not starting with the message from the second sentence and something like "High accuracy AM-technologies, like 2PP, are mainly limited to photo-curable polymers and currently lacks the possibility to produce multimaterial components. We present "

6. Line 65-67: Why do you call the process "phishing"? Doesn't the nanoparticles coagulation at the scaffolds come from the massively enlarged surface-volume-fraction of the micro/nanostructures compared to the substrate?

7. Line 137-138: The difference in wettability of the 2PP-printed scaffolds can be attributed to the difference in surface tension of water and isopropanol plus the probably non-polar characteristic of the

commercial resin. Are the contact angles in Supplementary Fig. 3 on the pure polymer film or modified with silane?

8. Line 183-186: How was the Au layer thickness measured?

9. Line 213-214: Did you also anneal the QD-coated structures and if yes, can the functionality of those be preserved?

Response to Reviewers' comments

We express our gratitude to the reviewers for providing us with their insightful and constructive feedback on our manuscript. We have modified and revised our manuscript based on the reviewers's comments and suggestions, and hope to have adequately addressed all the questions and concerns raised by the reviewers. We believe that these revisions have substantially improved the clarity of our work and highlighted its scientific contributions. To facilitate the reviewers' access to our responses, we have provided point-by-point replies to their comments, which are indicated by blue arrows (→) below. Additionally, we have listed all changes made in either the main text or Supplementary Information files of our manuscript and highlighted them in yellow.

Reviewer #1:

Summary:

The manuscript describes a strategy to address the materials limitation in 3D printing, harnessing TPP for 3D printing and multiple “immersing and retracting” process to coat nanoparticle layers on the 3D skeletons. The authors demonstrated the manufacturing of diverse materials, heterostructures, and the mixture. They also investigated the thermal treatment on the sintering of nanoparticles and removal of polymers. They provided an application example on 3D microrobot using magnetic nanoparticles for demonstration. I have some other comments on the papers, which should be well addressed before the paper can be accepted for publication.

→We appreciate the reviewer's accurate summary and kind suggestions. We have thoroughly revised our manuscript according to the reviewer's constructive suggestions, and believe its quality has been significantly improved.

Comment #1:

The 3D printing of inorganic materials is important for various device applications. However, using 3D-printed polymers as skeletons for further deposition/adsorption has been reported in the past one/two decades, see a recent paper in Nature 612, 685–690 (2022). Using “capillary trapping” is definitely different from previous strategies such as PVD, chemical coating, electrochemical coating, or adsorption,

while the advantage and drawback should be clearly elaborated in the paper. There are also some papers demonstrating direct laser 3D printing of colloidal nanocrystals, such as Science 377, 1112-1116 (2022); Science 381, 1468-1474 (2023) (refs 21 and 30 cited in the paper). While the authors said that the printing throughput is limited in the refs, how about other key parameters such as printing resolution, mechanical properties, ..., compared with the developed techniques. Moreover, I also spot that the discussion on ref. 21 is inaccurate in the manuscript, as no photoresist or monomer was used in that paper. Overall, the careful discussion on the printing capability of both the developed technique and other approaches should be done.

→ We thank the reviewer for the constructive suggestion. To provide a comprehensive comparison of the printing capabilities between our newly developed technique and existing methods, we have revised both the introduction and conclusion sections of the main text. Additionally, we included a detailed table in the supplementary materials, summarizing and contrasting the features of our approach with those of other technologies. Besides, regarding the colloidal nanocrystal ink (ref. 21), it is developed by grafting cross-linkable ligands onto the surface of nanocrystals, which enables direct-laser-writing of various 3D architectures. Although this ink differs from traditional monomers, we believe it still fits within the broader category of photoresists. Accordingly, we have clarified the discussion around ref. 21 in the revised manuscript.

In the main text:

(Page 2, line 41) Two strategies have been reported to 3D print non-polymeric materials, such as metals²⁰, quantum dots²¹⁻²³, metal oxides^{24,25}, and ceramics²⁶⁻²⁸ at the micro/nanoscale. One strategy relies on the direct-laser-writing of novel photoresists, which can either include designing cross-linkable precursors for targeted materials^{20,21,25,27-29}, or creating blends by physically dispersing nanomaterials into accessible monomers^{26,30}. After printing, the desired materials are either chemically bonded to other heterogeneous materials or physically embedded within a polymer matrix. To achieve pure 3D structures of the targeted material, post-treatments, such as pyrolysis and etching are necessary to remove the bonding materials^{20,26,29}. For example, a colloidal nanocrystal photoresist is developed by grafting cross-linkable ligands onto the nanocrystal surface, enabling the direct-laser-writing of various 3D architectures with a high nanocrystal mass content of ~90%²⁹. However, this method

is generally limited to specific chemistries, such as the unique surface ligands (C–H²⁹, Zn–S²¹ bonds) on colloidal nanocrystals and cross-linkable metal coordination compounds as certain metal precursors, hindering its broader applicability to other different materials. Besides, for photoresists composed of physically blended nanomaterials within accessible monomers, strict criteria of the nanomaterials need to be met, such as the size, mass loading, and transparency of the blended nanomaterials^{26,30}.

Another strategy to fabricate non-polymeric 3D micro/nanostructures involves template-assisted fabrication, which uses 3D-printed polymer scaffolds as templates for the deposition/adsorption of desired materials³¹. Techniques such as chemical coating³², physical vapor deposition⁸, atomic layer deposition³³⁻³⁵, and weak interaction-based adsorption (electrostatic force³⁶ or hydrogen bonding³⁷), have been used to deposit various inorganic materials onto these templates. These methods have produced inorganic material-coated structures with high resolution (~100 nm) and decent mechanical strength. However, they still face the challenges, such as the limited applicability to other materials, low mass loading of the desired materials, and the difficulties of integrating multiple materials. Detailed comparisons of the printing capabilities of existing approaches are provided in Supplementary Table 1. Despite these advancements, developing a versatile 3D micro/nanofabrication strategy that broadly accommodates wide-spectrum of materials (both polymeric and non-polymeric materials), achieves high material content, and facilitates the on-demand integration of multiple materials remains a significant challenge.

In the Supplementary Information:

Supplementary Table 1. Comparison among different micro/nanofabrication technologies of multi-material.

		Resolution	Roughness	Mechanical performance [#]	Mass loading of desired material	Material versatility	Printing speed	Distribution of desired material after printing	Ref.
Direct-laser writing	Cross-linkable metal coordinated compounds,	Sub-200 nm	/	~20 MPa, after photoresist removed by sintering ¹	<50%	Limited by specific chemistries	1 -50,000 μm/s	Chemically bonded with other heterogeneous materials	1,2
	Cross-linkable colloidal nanocrystals		~12 nm ³	>2 GPa, sintered at 700°C ⁴	~90%	Colloids (size< 30 nm) with specific surface ligands, e.g. C-H, Zn-S	1 - 50 μm/s	Embedded in the polymer matrix	3,4
	Physically blending with nanomaterials		/	4.5 GPa, sintered at 1200 °C ⁵	<50%	Transparent nanomaterial with size< 30 nm (silica and some ceramics)	1 -50,000 μm/s	Assembly of desired nanocrystals	5,6
Template-assisted fabrication	Adsorption via hydrogen bonding	~ 20 nm	~5 nm	/	~60%	Water dispensable nanomaterials	Up to 100,000 μm/s for the template printing, the deposition time of desired materials ranges from minutes to a few hours	Embedded in the polymer matrix	7
	Electrostatic adsorption	~ 100 nm*	/	/	Single nanoparticle layer with loosely discrete distribution	Limited by the surface group of nanomaterials, whose surface charge must be opposite to the template		8	
	Chemical coating		/	/	Depends on the amount of chemically active position of template surface	Limited by specific chemical reactions, e.g. electroless plating of metals		9	
	Atomic layer deposition		~1 nm ^{10,11}	0.28-2.56 GPa, after polymer removed ¹²⁻¹⁴	Accumulate along the deposition time	Metal, metal oxide, nitride, etc.		12-14	
	Physical deposition (sputtering, evaporating deposition, etc.)		~1 nm ¹⁵	/		Metal, metal oxide		16	
	Capillary trapping assisted deposition (this work)		~46 nm	Material- and structure-dependent	Accumulate along the immersing/retracting cycles	Broader material applicability (polymeric and non-polymeric nanomaterials) (size: 5-500 nm, Without considering specific chemistries)			

#: Engineering compressive strength.

/: No related data reported in literatures.

*: It depends on the resolution of the 2PP-printed scaffold, which is around 100 nm¹⁷

Comment #2:

The interparticle van der Waals force is basically a weak force. What is the mechanical properties of the fabricated 3D structures?

→ We thank the reviewer's constructive suggestions, and have conducted additional measurements for the mechanical properties of the fabricated microstructures. Yes, we do agree that interparticle van der Waals force is basically a weak force, while the thermal sintering process can enhance the cohesion among these particles as high temperatures can improve their interfacial bonding (such as melting). Accordingly, we added these experimental results and discussions in the revised manuscript.

In the main text:

(Page 7, line 247)

Mechanical properties of the as-fabricated structures

The structural stability of our 3D microstructures can be significantly enhanced through thermal annealing (Fig. 4), which shows the mechanical performance of nanomaterial-assembled microstructures before and after thermal annealing at different temperatures during compression tests. Initially, these structures are primarily stabilized by van der Waals forces, which are less robust than covalent or ionic bonds²⁹. To enhance their interaction and strength, we implement a post-annealing process. Given that nanomaterials typically have lower melting points than their bulk counterparts, we can achieve stronger interfacial bonds through low-temperature sintering^{29,37,51}. This process is feasible because the IPS polymeric scaffolds we employ are thermally stable below 300 °C. The interfacial melting of the nanoparticles during annealing remarkably enhances the mechanical strength of the structures. For example, an IPS microcube assembled with Au NPs was annealed at 120 °C and 240 °C, causing the Au NPs to partially melt and reconfigure into a dense, continuous 3D porous Au upon cooling (Fig. 4a). Remarkably, both samples annealed at these temperatures (with a density of ~ 4.18 g/cm³) exhibited a higher compressive strength than their untreated counterpart (Fig. 4b) with enhancements of 160% and 120% in their maximum compressive strength for the respective temperatures. However, it is important to note that higher sintering temperatures do not invariably result in

increased mechanical strength; excessive melting of Au nanoparticles can lead to uneven nanoparticle coagulation or potential degradation of the IPS scaffold.

When the annealing temperature exceeds the degradation temperature of polymer scaffolds, the polymer can be completely removed, resulting in a 3D microstructure composed solely of the targeted materials. For instance, Fig. 4c show the morphological comparison of a 3D fullerene-like microsphere assembled with Fe_3O_4 NPs before and after annealing at 600 °C for 2 hours. The annealed structure maintains good shape integrity with slight shrinkage and deformation, despite the removal of the polymer scaffold. The key to preserving the high integrity of these annealed microstructures at elevated temperatures is ensuring a uniform and high loading of nanoparticles on the micro-scaffolds. This uniformity is crucial as non-uniform or sparse distribution of nanomaterials can lead to structural instability, especially due to the thermal degradation of the IPS polymer scaffold. A typical example of this issue is observed in the same type of 3D fullerene-like microsphere, which collapses after annealing at 600 °C due to a lesser load of Fe_3O_4 nanoparticles (Supplementary Fig. 16). Once the polymer template is removed, the pure materials exhibit increased rigidity; this is evidenced by the steeper force profiles observed at higher annealing temperatures (Fig. 4d).

Fig. 4 Mechanical performance of nanomaterial-assembled microstructures. a, SEM images of Au NPs annealed at different temperatures. **b,** Compressive force-

displacement curves of Au NP-assembled cube-shaped microscaffolds before and after annealing at 120 °C, 240 °C. Insets are SEM images show the morphology of an Au NPs/IPS micro-scaffold before (left) and after compression (right). **c**, SEM images of Fe₃O₄ NP-assembled beams of a 3D hollow microsphere before (left) and after annealing at 600 °C (middle), where the cross-section SEM image (right) from FIB-cut beam shows the polymer scaffold is removed. **d**, Compressive force-displacement curves of Fe₃O₄ NP-assembled microspheres before and after annealing at 300 °C, 600 °C, and 900 °C. Insets are SEM images showing the morphology of a 900 °C annealed Fe₃O₄ NPs microsphere before and after compression. Scale bars are 200 nm in **(a)**, 60 μm in **(b, d)**, 30 μm in **(c, left and middle)**, 1 μm in **(c, right)**.

Comment #3:

Without sintering, how the weak van der Waals force overcomes the capillary force during evaporation?

→We thank the reviewer for raising this question. While van der Waals forces are typically considered weak, our results indicate they are sufficiently strong to overcome capillary forces during the evaporation and subsequent deposition cycles. To substantiate this claim, we conducted additional experiments to track the deposition of nanoparticles at various stages of the deposition cycles. As illustrated in Supplementary Fig. 2, nanoparticles consistently accumulate and adhere to the microscaffolds, including a hollow microsphere (Supplementary Fig. 2A) and a microcube (Supplementary Fig. 2B), through the cyclic deposition (immersion and retraction). This steady accumulation clearly shows that the interparticle van der Waals forces are robust enough to counteract the capillary forces during ongoing deposition cycles. Furthermore, we immersed the nanoparticle-deposited scaffolds in a pure solvent to assess if the capillary forces during evaporation could dislodge nanoparticles from the scaffolds (Supplementary Fig. 3). Remarkably, even after 600 evaporation cycles, there was no noticeable change in the deposition morphology on the scaffolds. These findings lead us to conclude that although van der Waals forces are generally weak, they are adequately strong to overcome capillary forces, particularly given that nanoparticles possess higher surface areas which has a high interaction strength. These results and the corresponding discussion have been added to the revised manuscript.

In the main text:

(Page 4, line 123) The steady accumulation (Supplementary Fig. 2) indicates that the interparticle van der Waals forces are robust enough to counteract the capillary forces during ongoing deposition cycles. Furthermore, we immersed the as-assembled scaffolds in a pure solvent to assess if the capillary forces during evaporation could dislodge nanoparticles from the scaffolds (Supplementary Fig. 3). Remarkably, even after 600 evaporation cycles, there was no noticeable change in the deposition morphology on the scaffolds. These findings lead us to conclude that although van der Waals forces are generally weak, they are adequately strong to overcome the capillary forces, particularly given that nanoparticles possess high surface areas, enabling a high interparticle adhesion strength.

In the Supplementary Information:

Supplementary Fig. 2 (A, B) SEM images of a hollow microsphere assembled with BaTiO₃ (A) and a cube-shaped micro-scaffold assembled with Fe₃O₄ (B) for different immersing/retracting cycles.

Supplementary Fig. 3 SEM images of Au NPs assembled microsc scaffold before and after 600 immersing/retracting cycles in pure 90% IPA/H₂O solution (no nanoparticles contained). The morphology basically remains unchanged on the structure after immersing, indicating that the assembled particles are robust enough to resist the redispersion in the following immersing process.

Comment #4:

What is the resolution of this approach? What is the minimum thickness of the coated nanoparticle layer? What is the density of the fabricated structures? What is the surface roughness? These key factors are important for the future application of the technique.

→ We thank the reviewer for the constructive suggestion. In response, we have conducted additional experiments and calculations. We observed that IPS line scaffolds with deposited Au nanoparticles (40 nm) display controllable widths at the submicron scale, with the narrowest being around 400 nm (Supplementary Fig. 5). We also measured the average surface roughness (R_a) of the deposited Au nanoparticle layer using an atomic force microscope, which is $46.6 \text{ nm} \pm 3.9 \text{ nm}$. The density of the deposited scaffolds is calculated to be around 4.18 g/cm^3 . Furthermore, regarding the minimum thickness achievable, it is dependent on the particle size used for deposition.

Theoretically, the minimum thickness should be equal to the diameter of the particles if a single layer of nanoparticles is uniformly deposited onto the microstructure. However, achieving a uniform single-layer deposition of nanoparticles is challenging, and multilayer deposition is more common. We have included these data and observations in the revised manuscript.

In the main text:

(Page 4, line 147) it can assemble nanomaterials onto micro/nanostructures with a submicron resolution (Supplementary Fig. 5) and a roughness of $46.6 \text{ nm} \pm 3.9 \text{ nm}$ (Supplementary Fig. 6).

(Page 7, line 260) Remarkably, both samples annealed at these temperatures (with a density of $\sim 4.18 \text{ g/cm}^3$) exhibited a higher compressive strength than their untreated counterpart (Fig. 4b) with enhancements of 160% and 120% in their maximum compressive strength for the respective temperatures.

(Page 14, Line 579) **Estimation of the density of the fabricated structures.** To obtain the density of nanomaterial-assembled microstructures, a cube-shaped micro-scaffold assembled with Au NPs after 600 immersing/retracting cycles is taken as an example. The thickness of the assembled Au NP layer is $\sim 0.32 \text{ }\mu\text{m}$. The density of IPS polymer (ρ_{IPS}) is 1.19 g/cm^3 , and the density of Au (ρ_{Au}) is 19.32 g/cm^3 . The volumes of a pure IPS cube-shaped micro-scaffold (V_{IPS}), an Au NP-assembled micro-scaffold (V_t), and the assembled Au NP layer (V_{Au}) can be calculated from the cube model and measurement, and the density of the assembled cube can thus be estimated as:

$$\rho = \frac{\rho_{IPS} \cdot V_{IPS} + \rho_{Au} \cdot V_{Au}}{V_t} \approx 4.18 \text{ g/cm}^3$$

In the Supplementary Information:

Supplementary Fig. 5 SEM image of Au NP-assembled IPS line scaffolds with different widths. As the strategy is based on template-assisted fabrication, the resolution of this method thus depends on that of the 2PP-printed scaffolds (which could be further down to ~100 nm).

Supplementary Fig. 6 Roughness characterization of nanomaterial-assembled structures. (A, B) AFM measurement (A) and SEM image (B) of an Au NPs (40 nm) assembled surface after 900 immersing/retracting cycles.

Comment #5:

The coating of nanoparticles on the skeletons relies on the wettability. That means the surface chemistry of both the nanoparticles and the 2PP-printed 3D skeletons, as well as the solvent, should be important. I expect more discussion here.

→We thank the reviewer for the insightful suggestions. We agree with the reviewer that the surface chemistry of both the nanoparticles and the 2PP-printed 3D skeletons, as well as the choice of solvent, play pivotal roles in achieving uniform and efficient deposition of nanoparticles on the skeletons. Uniform dispersion of nanomaterials within the solvent is the prerequisite to ensure their later uniform deposition on the micro-scaffolds, and the solution's high affinity to the 3D micro-scaffolds matters most. Furthermore, the selective affinity of both the substrate and polymer micro-scaffolds towards the solvent is also crucial for the efficient deposition. In short, beyond the rational design of solvents for deposition (Supplementary Fig. 8), surface treatments such as O₂ plasma treatment, chemical modification, or vapor deposition of nanomaterials, micro-scaffolds, and substrates can significantly enhance deposition efficiency and uniformity. These techniques adjust their mutual selective affinity (wettability), improving their interaction dynamics. Detailed discussions and experimental results have been added to the revised manuscript to further elucidate these points.

In the main text:

(Page 4, line 158) Uniformly dispersed nanomaterials in the solvent is the prerequisite to ensure their uniform deposition onto the micro-scaffolds, and the most critical factor is the solution's high affinity to the 3D micro-scaffolds.

(Page 5, line 178) In short, beyond the rational design of solvent used for the deposition (Supplementary Fig. 8), surface treatments of nanomaterials, micro-scaffolds, and substrates, including O₂ plasma treatment, chemical modification, or vapor deposition, can also enhance the efficiency and uniformity of the deposition, as these techniques can adjust their mutual selective affinity (wettability).

In the Supplementary Information:

Supplementary Fig. 8 SEM images of a honeycomb-like micro-scaffold assembled with Fe₃O₄ NPs after 1600 immersion-retraction cycles in toluene. Different solvent systems, such as the non-polar solvents, can be employed to achieve similar results.

Comment #6:

The versatility of the proposed techniques should be carefully backup. For instance, how to make a volumetric 3D structures with high purity? It should be clearly pointed out what can be done and what cannot be achieved, as we know no technique is perfect.

→We appreciate the reviewer's constructive suggestion. Accordingly, we have expanded the discussion on the limitations of our techniques, which we had not fully addressed in the previous version of the manuscript. Specifically, regarding the capability of our techniques to create volumetric 3D structures, we conducted additional experiments (Supplementary Fig. 2). We observed that as deposition cycles increase, the pores within the structures gradually become blocked by the deposited nanoparticles, potentially leading to hollow interiors. Consequently, it is challenging to achieve fully volumetric 3D structures using our method. This limitation, among others, is now thoroughly discussed in the revised conclusions and Supplementary Information.

In the main text:

(Page 8, line 316) Although we have demonstrated the on-demand fabrication of versatile materials into 3D micro/nanostructures, several challenges remain for our future research. For instance, it has been observed that these micro/nanostructures

are often composed of a relatively rough surface and random assembly of nanoparticles, which could potentially result in inferior properties compared to their bulk counterparts. Additionally, limited to the mechanism of the capillary capture of nanomaterials, our strategy primarily facilitates the creation of 3D scaffolds rather than achieving true volumetric 3D structures (see Supplementary Fig. 2). Furthermore, the exploration of milder etching technologies, such as plasma and wet etching, is necessary to extend the material applicability for fabricating pure-material microstructures (as some material would lose their functionality at high temperatures, e.g. QDs^{54,55}), improve the morphology and structural integrity of the pure-material micro/nanostructures.

In the supporting information:

Supplementary Fig. 2 (A, B) SEM images of a hollow microsphere assembled with BaTiO₃ (A) and a cube-shaped micro-scaffold assembled with Fe₃O₄ (B) for different immersing/retracting cycles.

Comment #7:

I can spot some defects generated in Fig. 4 after annealing at 600-700 °C. However, I did not see discussion on the structural change and the impact on mechanical properties and applications.

→We thank the reviewer's kind suggestion. We have added the discussions on the structural change and mechanical properties of the fabricated structures before and after annealing in the revised text.

In the main text:

(Page 7, line 266) When the annealing temperature exceeds the degradation temperature of polymer scaffolds, the polymer can be completely removed, resulting in a 3D microstructure composed solely of the targeted materials. For instance, Fig. 4c show the morphological comparison of a 3D fullerene-like microsphere assembled with Fe₃O₄ NPs before and after annealing at 600 °C for 2 hours. The annealed structure maintains good shape integrity with slight shrinkage and deformation, despite the removal of the polymer scaffold. The key to preserving the high integrity of these annealed microstructures at elevated temperatures is ensuring a uniform and high loading of nanoparticles on the micro-scaffolds. This uniformity is crucial as non-uniform or sparse distribution of nanomaterials can lead to structural instability, especially due to the thermal degradation of the IPS polymer scaffold. A typical example of this issue is observed in the same type of 3D fullerene-like microsphere, which collapses after annealing at 600 °C due to a lesser load of Fe₃O₄ nanoparticles (Supplementary Fig. 16). Once the polymer template is removed, the pure materials exhibit increased rigidity; this is evidenced by the steeper force profiles observed at higher annealing temperatures (Fig. 4d).

Reviewer #2:

Summary:

The manuscript titled "Capillary trapping of wide-spectrum nanomaterials for three-dimensional micro/nanofabrication" by Lyu et al. describes a processing route for multi-material deposition of colloidal systems on printed 2PP frameworks/scaffolds. The work is based on the vision that the proposed processing route offers a new technology for forming multi-material components in a size range that cannot be achieved by methods other than 2PP.

→ We appreciate the reviewer's careful evaluation of our work, and the reviewer's constructive comments/suggestions have helped to improve the quality of our manuscript. To better clarify the contribution and further strength the innovation of our work, we have summarized the innovative aspects of our work into the following **key points (1-3)**:

1. This is the first instance of utilizing capillary force-mediated assembly for general 3D micro/nanofabrication with non-specific materials. While other interactions (chemical bonds, Science, 2023, 381, 1468; size-dependent steric effects, Science, 2022, 378, 1325) have aided 3D construction, they often impose stringent criteria on the printability of nanomaterials, such as necessary the surface ligands (abundant C-H bonds), and size restrictions (typically less than 100 nm). Capillary force, which is dominant during micro/nanoscale evaporation, offers broad adaptability to materials regardless of their specific properties like type, size, shape, surface charge, or functional groups.

2. 3D microstructures composed of pure materials can be obtained. High loading and uniformity of nanomaterials are achieved using the salting-out effect, which maintains structural integrity even after the removal of the polymer micro-scaffolds. Compared other techniques that show low material loadings (Science, 2018, 362, 1281; Science, 2022, 378, 1325), fabricating self-supporting 3D microstructures solely from pure materials without significant deformation or collapse presents significant challenges.

3. Our strategy excels in integrating multiple materials into a single microstructure. It is straightforward to produce a microstructure with homogeneously distributed materials, such as the co-deposited Au/Ag (Science, 2022, 378, 1325) and Cu/Ni (Nature, 2022, 612(7941): 685-690) alloy structure. However, fabricating heterogeneously distributed materials within a microstructure has been challenging. our method facilitates easy integration of multiple materials, especially in a layer-by-layer fashion. This capability has significant potential for fabrication of heterogeneous, multifunctional micro/nanodevices tailored for specific, synergetic, and on-demand functionalities.

Overall, the main contribution of this work is that we proposed a novel 3D micro/nanofabrication strategy with non-specific materials, ranging from polymer to metal, metal oxide, quantum dots, and so on. Within the clarification of the concept and further elaboration in the revised manuscript, we believe that our work represents a significant contribution to the field.

Comment #1:

To support this opinion, it is stated that 2PP is not capable of processing materials other than polymers. In this context, it states (line 54): “On the other hand, the materials printed through these chemical or physical strategies are either chemically bonded with organic photoresist initiators or embedded in a polymer matrix, which can limit the functionality of the resulting microstructures. Developing a highly efficient 3D micro/nanofabrication strategy that is compatible with any non-specific materials, such as polymeric and non-polymeric materials, remains a significant challenge.” In fact, this statement is not true. ref 28 essentially follows a conventional ceramic processing route and delivers 3D microstructures of high-performance ceramics after sintering. This process and the products are no different from continuously produced ceramics, in fact they are better as the structural features are smaller than the critical defect size. In this sense, I do not see the argument here that the 2PP molded ceramics are inferior to standard high-performance ceramics.

→We thank the reviewer for the constructive comment. Beyond direct-laser-writing of novel photoresists, creating blends by physically dispersing nanomaterials into accessible monomers as suggested by the reviewer provides another general strategy

for 3D microstructures from non-polymeric materials. This approach allows to create ceramics with properties comparable to their bulk counterparts following a high-temperature annealing process (previously Ref. 28). However, it typically imposes strict criteria on the design of the nanomaterials used in the ink, such as size, mass loading, and transparency of the blended nanomaterials. Moreover, this strategy is generally limited to inorganic materials that can withstand high-temperature annealing as polymer matrix has to be removed, thus narrowing its applicability in other materials. We have revised the introductory sections of our manuscript to thoroughly discuss both the advantages and limitations of this method.

In the main text:

(Page 2, line 41) Two strategies have been reported to 3D print non-polymeric materials, such as metals²⁰, quantum dots²¹⁻²³, metal oxides^{24,25}, and ceramics²⁶⁻²⁸ at the micro/nanoscale. One strategy relies on the direct-laser-writing of novel photoresists, which can either include designing cross-linkable precursors for targeted materials^{20,21,25,27-29}, or creating blends by physically dispersing nanomaterials into accessible monomers^{26,30}. After printing, the desired materials are either chemically bonded to other heterogeneous materials or physically embedded within a polymer matrix. To achieve pure 3D structures of the targeted material, post-treatments, such as pyrolysis and etching are necessary to remove the bonding materials^{20,26,29}. For example, a colloidal nanocrystal photoresist is developed by grafting cross-linkable ligands onto the nanocrystal surface, enabling the direct-laser-writing of various 3D architectures with a high nanocrystal mass content of ~90%²⁹. However, this method is generally limited to specific chemistries, such as the unique surface ligands (C-H²⁹, Zn-S²¹ bonds) on colloidal nanocrystals and cross-linkable metal coordination compounds as certain metal precursors, hindering its broader applicability to other different materials. Besides, for photoresists composed of physically blended nanomaterials within accessible monomers, strict criteria of the nanomaterials need to be met, such as the size, mass loading, and transparency of the blended nanomaterials^{26,30}.

Comment #2:

Furthermore, polymer-derived ceramics and sol-gel technologies in combination with 2PP enable the production of advanced functional materials, as has been proven by a considerable body of literature devoted to this topic. Furthermore, I do not see that the impregnation of frameworks/scaffolds is new. Impregnation of foams is a widely used technology that is very similar to the methodology presented, except that the foams are not 2PP molded. After firing/sintering the foams disappear. On this particular point, I admit that the work represents a new approach, as it uses 2PP for the manufacture of scaffolds, however, the results obtained are not convincing and are at a preliminary stage. Unlike existing technologies, which generally follow the same strategy, the parts presented in this study do not have advanced characteristics.

→We thank the reviewer for recognizing our approach as a novel strategy. After a thorough review of the literature, we acknowledge that the impregnation of frameworks/scaffolds has indeed been reported in some studies (Nature, 2021, 595(7865): 58-65; Small, 2023, 19(22): 2207822), while these works primarily focus on the macroscale (above centimeter level). Systematic studies of capillary interactions at smaller scales, particularly at the micro- and nanoscale, remain sparse, and it is unclear whether these dominant physical interactions can effectively scale down to such diminutive dimensions. Furthermore, there is a lack of research demonstrating how to efficiently, uniformly, and selectively deposit nanomaterials onto micro-scaffolds rather than onto substrates, a significant technical challenge given the difficulties involved in manipulating materials at these small scales.

Our strategy not only demonstrates the feasibility of using capillary trapping to achieve steady 3D accumulation of various nanoparticles but also offers a method to integrate multiple materials into a single microstructure with on-demanded functionalities, such as magnetic responsive microrobots with piezoelectric performance. Besides, we have rewritten the introductory sections and included a detailed table in the supplementary information, summarizing and contrasting the features of our approach with those of other technologies to help enable a more comprehensive understanding of our approach. Please refer to the reply to the **Comment #1** of **Reviewer #1**. Within the clarification of the concept and the revision, we hope that the reviewer finds the revised version of our manuscript to be an improved and informative contribution to the field.

Reviewer #3:

Summary:

The work described in the manuscript addresses an original approach for nanoparticle assembly using capillary forces in a 3D printed micro scaffold. Overall this is great work and truly new, and certainly forward looking with new opportunities for nanomaterial assembly into functional constructs. The authors have asked the key scientific questions and addressed many issues that can be expected, but well addressed, or unforeseen and then smartly studied for their influence on the and also addressed.

The strength of the paper is its quite simple but strong and generic concept idea, and the execution of a battery of parameter screening. The authors have used the now well-established 2PP for scaffold printing, and then assembled a multitude of mono- or up to 4 heterogenous material in a sequential manner. Process was optimized to increase assembly control and yield by solvent modification for contact angle tuning, and fishing/drying timing was also studied systematically for optimal layer growth, as well as adding salt and post assembly annealing. The grown films were then analyzed with fluorescent, TEM and other microscopic techniques to quantify, sometime semi-quantitatively, the quality and metrology of the NP assembly. First order validation of fabricated devices includes the motion under an electric field of magnetic nanorobots made with the 2PP-assembly method.

The text reads well, and the figures are informative and concise. The idea of the work clearly qualifies for a paper in this journal, but would need a few fixes and add-ons that seem to be missing, and that would complete the study for a larger readership.

→We appreciate the reviewer's accurate summary and positive opinion about our work. We have revised our manuscript according to the reviewer's constructive suggestions, and believe its quality is much improved.

Comment #1:

The work is based on 2PP made scaffolds. In most cases the printed scaffold remains under the assembly, or is removed by the high temperature sintering. This means that the costly scaffold can only be used once. Is there no way to preserve the scaffold for

multiple assemblies, so that this opens a route for mass manufacturing of the built nanosystems?

→ We appreciate the reviewer for the kind suggestion. Indeed, it is possible to preserve the printed scaffold while removing assembled nanomaterials. For instance, a sacrificial layer could be applied prior to nanomaterial deposition for the scaffold recycles. This layer can be prepared using methods such as spin-coating a layer of PMMA, sputtering a layer of aluminum, or employing other chemical surface modifications. Following the deposition of nanomaterials, the sacrificial layer can be dissolved using appropriate etchants, such as acetone or a NaOH solution. This process allows the assembled nanomaterials to redisperse in the solution while leaving the printed scaffold intact. We have incorporated further discussion on this technique in the revised conclusions of our manuscript.

In the main text:

(Page 12, line 499) In addition, a sacrificial layer could be applied prior to nanomaterial deposition for the scaffold recycles, and dissolving the layer could allow the assembled nanomaterials to release in the solution and thus recycle the scaffolds.

Comment #2:

some immersion-fishing cycles go up to several hundreds. What is the time this needs to be done. Maybe the time scale could be more clearly highlighted.

→ We appreciate the reviewer for this question. In our current work, a typical immersing/retracting cycle, essential for the drying process, takes about 80 seconds. This duration includes a 50-second waiting period, necessary for the complete evaporation of the solution trapped within the micro-scaffolds, with the physical movement up and down accounting for the remaining 30 seconds. The complexity of the microstructures directly influences the length of the waiting time, as more intricate structures require longer periods to ensure full evaporation. To enhance the efficiency of the drying process, methods such as increasing air circulation around the microstructures with a small fan, or applying gentle heat using infrared light, can be employed, which can significantly reduce the drying time to just a few seconds. These

modifications and their impact on the drying process have been discussed further in the revised manuscript.

In the main text :

(Page 6, line 211) Given that the throughput of this approach is significantly influenced by the number of immersion-retraction cycles, several strategies can be implemented to reduce the deposition time. First, increasing the concentration of nanoparticles in the solution can reduce the required number of cycles. Second, the drying time of the trapped solution during each cycle can be drastically shortened (from the current 50 seconds to less than 10 seconds) by implementing measures such as reducing humidity, enhancing air ventilation, and raising the temperature with infrared light.

Comment #3:

It is not entirely clear what the purpose of the BaTiO₃ buckyball is. The piezoresponse results are not well described and leaves the reader a bit alone.

→We thank the reviewer for this question. The BaTiO₃ contributes to the generation of piezoelectric voltage when pressure (from focused ultrasound) is applied to them, thus it can be potentially used for targeted electrical stimulation. To make it clearer, we have removed original vague description about the BaTiO₃ buckyball and add discussion in the revised manuscript.

In the main text

(Page 8, line 281) Heterogeneous 3D microstructures can be fabricated by co-deposition multiple nanomaterials.

(Page 8, line 284) In addition, this technique is allowing for the creation of the micro/nanodevices with diverse materials, equipped with on-demand functionalities.

(Page 8, line 289) These annealed Fe₃O₄-BaTiO₃ microrobots can be magnetically driven (contributed from Fe₃O₄) and manipulated as demonstrated in Fig. 5k and Supplementary Video 3, and exhibit piezoelectric property (contributed from BaTiO₃) as well. The annealing process doesn't notably degrade the piezoelectric performance as the piezoelectric current is ~8.0 pA (originally ~9.5 pA) under the treatment of

focused ultrasound (FUS) with a power of $100 \text{ mW}\cdot\text{cm}^{-2}$ (Fig. 5I). These multifunctional microrobots can be wirelessly manipulated and used for targeted electrical stimulation, demonstrating significant potential for biomedical applications, particularly in targeted cell stimulation⁵².

Comment #4:

The melting temperature of Au NP at 50 or 70nm is certainly higher than the used 240 degC, so this reviewer is not sure that this will lead to a melting of the particles. →Yes, we agree with the reviewer that the melting temperature of Au nanoparticles (40 nm used in these experiments) is typically higher than 240°C. In our experiments, while the temperatures were not sufficient to fully melt the Au nanoparticles, they were adequate for partial melting, particularly at the interfaces. This is evidenced by the roughened morphologies observed after annealing (Fig. R1) and the enhanced overall mechanical properties (Fig. 4). These observations suggest that interfacial melting and bonding are contributing to these changes. We have clarified these points and added further details in the revised manuscript to enhance understanding of the phenomena involved.

Figure R1. SEM images of Au NPs annealed at different temperatures

In the main text:

(Page 7, line 258) For example, an IPS microcube assembled with Au NPs was annealed at 120 °C and 240 °C, causing the Au NPs to partially melt and reconfigure into a dense, continuous 3D porous Au upon cooling (Fig. 4a).

Comment #5:

NP colloidal solutions need surfactants to prevent agglomeration. What happens with the surfactants after the assembly? Could it still be there and 'pollute' the material?
→ We thank the reviewer for this question. Indeed, a surfactant is typically required to ensure the proper dispersion of nanoparticle colloids in most cases. Although residual surfactant remains within the nanoparticles after assembly, it can largely be removed by immersing the assembled structure in deionized (DI) water for a few minutes (Supplementary Fig. 11). For commercially available nanoparticle suspensions, the surfactant concentration can be reduced by centrifuging the suspension and discarding the supernatant before redispersing the nanoparticles in an IPA/H₂O solvent mixture. We have included these procedural details in the revised manuscript to provide clarity on the surfactant removal process.

In the methods:

(Page 12, line 485) The initial nanomaterial suspension is first centrifuged, followed by discarding the supernatant to decrease the concentration of surfactants from the original solution.

Comment #6:

what is the authors observation of assembled NP from a previous step are eventually repositioned or removed in a subsequent step, knowing that capillary forces are strong.

→ We thank the reviewer for raising this insight suggestion. While van der Waals forces are typically considered weak, our results indicate they are sufficiently strong to overcome capillary forces during the evaporation and subsequent deposition cycles. For more details please refer to the reply to the **Comment #3 of reviewer #1**.

Comment #7:

If NP are uniform in size distribution, they can form crystalline features. Any comment or observation? How does the 2PP 3D printed scaffold influences crystal like assembly?

→ We thank the reviewer for this interesting point. Using our strategy, we found it is challenging to achieve perfect crystalline features of these NPs. Despite the uniform size and shape of the Au nanoparticles, Au nanorods, and PS microparticles in our experiments, only short-range ordered assemblies of the PS spheres (500 nm) were

observed, typically when a single layer is formed, as depicted in Fig. R2. Achieving long-range crystalline features has proven to be generally difficult, as illustrated in Fig. R3. We have included additional discussion on these challenges and observations in the revised manuscript.

Fig. R2 SEM images of a microstructure assembled with PS microsphere (500 nm) of a low content.

Fig. R3 SEM images showing the morphologies of the Au NPs, Au nanorods, PS sphere on microstructures after multiple immersing/retracting cycles.

In the main text:

(Page 8, line 317) For instance, it has been observed that these micro/nanostructures are often composed of a relatively rough surface and random assembly of nanoparticles, which could potentially result in inferior properties compared to their bulk counterparts.

Comment #8:

Is there a way to speed up the drying step in between the dipping steps, e.g. by some gentle heating or would it compromise the dynamics of the drying process for the assembly?

→We appreciate the reviewer for the kind suggestion. To enhance the efficiency of the drying process, methods such as increasing air circulation around the microstructures with a small fan, or applying gentle heat using infrared light, can be employed, which can significantly speed up the drying step and reduce the drying time, while it doesn't bring about any noticeable compromise to the dynamics of the drying process.

In the main text:

(Page 6, line 211) Given that the throughput of this approach is significantly influenced by the number of immersion-retraction cycles, several strategies can be implemented to reduce the deposition time. First, increasing the concentration of nanoparticles in the solution can reduce the required number of cycles. Second, the drying time of the trapped solution during each cycle can be drastically shortened (from the current 50 seconds to less than 10 seconds) by implementing measures such as reducing humidity, enhancing air ventilation, and raising the temperature with infrared light.

Reviewer #4:**Summary:**

This manuscript demonstrates a versatile and easy to adapt process to coat 2PP-printed micro-and nanostructures with a variety of nanoparticles. The materials span from metal over low sintering temperature ceramics and polymers to quantum dots. The adjustment of the suspension for the scaffold wetting and the surface charges of the nanoparticles allows for a uniform coagulation of the particles at the structures during the drying process. The concept shows a simple way to produce even functional devices but comes with a few flaws and drawbacks, which must be discussed carefully.

→ We thank the reviewer for the careful evaluation of our work and kind suggestions. We have revised our manuscript according to the reviewer's constructive suggestions, and believe its quality is much improved.

Comment #1:

The process is a form of dip-coating on structures materials with particle assembly during drying. Why is the word "coating" first and only used in line 194?

→ We agree with the reviewer that the process is a form of dip-coating. The term "coating" is a more general expression. To be more specific, we would like to use the expression of 'assembly' to emphasize the capillary force-directed assembly process. To maintain the coherence through the article, we have revised corresponding places in the revised manuscript (Page 3, line 92; Page 4, line 143; Page 6, line 205, and so forth).

Comment #2:

The variety of materials, which can be coated, is impressive, but the authors should be careful. The resulting structures do not possess the same properties as if they would be completely consisting of either metal, ceramic or a polymer, as those are core-shell-structures.

→ We appreciate the reviewer's kind suggestion. We have clarified this and carefully summarized drawbacks of our method in the revised manuscript.

In the main text:

(Page 8, line 316) Although we have demonstrated the on-demand fabrication of versatile materials into 3D micro/nanostructures, several challenges remain for our future research. For instance, it has been observed that these micro/nanostructures are often composed of a relatively rough surface and random assembly of nanoparticles, which could potentially result in inferior properties compared to their bulk counterparts. Additionally, limited to the mechanism of the capillary capture of nanomaterials, our strategy primarily facilitates the creation of 3D scaffolds rather than achieving true volumetric 3D structures (see Supplementary Fig. 2). Furthermore, the exploration of milder etching technologies, such as plasma and wet etching, is necessary to extend the material applicability for fabricating pure-material microstructures (as some material would lose their functionality at high temperatures, e.g. QDs^{54,55}), improve the morphology and structural integrity of the pure-material micro/nanostructures.

Comment #3:

The authors should also discuss their results compared to differently coated and hollow 2PP-structures like:

- a. Formanek, F., Takeyasu, N., Tanaka, T., Chiyoda, K., Ishikawa, A., & Kawata, S. (2006). Three-dimensional fabrication of metallic nanostructures over large areas by two-photon polymerization. *Optics express*, 14(2), 800-809.
- b. Bauer, J., Hengsbach, S., Tesari, I., Schwaiger, R., & Kraft, O. (2014). High-strength cellular ceramic composites with 3D microarchitecture. *Proceedings of the National Academy of Sciences*, 111(7), 2453-2458.
- c. Meza, L. R., & Greer, J. R. (2014). Mechanical characterization of hollow ceramic nanolattices. *Journal of materials science*, 49, 2496-2508.

→We thank the reviewer's kind suggestion. We have comprehensively compared the printing capability of different methods in Supplementary Table 1. These works mentioned above are all cited and included in the table. For details please refer to the reply to the **Comment #1** of **reviewer #1**.

Comment #4:

The 2PP-printed scaffolds are from a commercial resin with unknown ingredients. Most likely they are silicon-based pre-ceramic polymers. During annealing the organic part indeed burns off but the glassy silicon oxide remains. There is no proof shown, that the used polymer/preceramic polymer is removed (EDS/EDX for Si?).

→We thank the reviewer for raising this issue. The IPS photoresist we used in this work is a methacrylate-based resin, which does not contain silicon-based pre-ceramic polymers. We did the EDS mapping for Si on a pure IPS scaffold, as shown in Fig. R4, the results showed strong Si signals from the silicon substrate, while no Si signal were detected on the polymer structure. After annealing at high temperature in the air atmosphere, the polymer scaffold can be fully removed. We have accordingly added the information in the revised manuscript.

Fig. R4 SEM image (left) of a pure IPS scaffold and the EDS mapping of Si element (right).

In the main text:

(Page 12, line 477) All 3D microstructures in this work are printed using a commercial direct-laser-writing system (Photonic Professional GT, Nanoscribe GmbH) with a 25x objective and IP-S photoresist (a methacrylate-based resin).

Comment #5:

Line 12-13: The introduction sentence “Fabrication of three-dimensional (3D) micro/nanostructures with wide-spectrum materials has been increasingly important across various fields.” comes in a little bumpy. The fabrication is still important, not

was and a “wide range of materials” and “across various fields” are rather generic phrases. Why not starting with the message from the second sentence and something like “High accuracy AM-technologies, like 2PP, are mainly limited to photo-curable polymers and currently lacks the possibility to produce multimaterial components. We present “

→We appreciate the reviewer for this kind suggestion. We agree with the reviewer and adopt the suggestion.

In the abstract:

(Page 1, line 14) High-precision additive manufacturing (AM) technologies, such as two-photon polymerization (2PP), are mainly limited to photo-curable polymers and currently lacks the possibility to produce multimaterial components.

Comment #6:

Line 65-67: Why do you call the process “phishing”? Doesn’t the nanoparticles coagulation at the scaffolds come from the massively enlarged surface-volume-fraction of the micro/nanostructures compared to the substrate?

→We thank the reviewer for raising this question. In our method, the fishing process can serve as an apt analogy. Initially, a mixture of water and fish is captured by the net, similar to how our technique employs capillary trapping. Following this, just as water flows out while the fish remain caught (akin to micro-scaffolds in our context), our deposition process results in the nanoparticles (comparable to the fish) being retained and assembled onto the scaffolds. We hope this clarification can effectively illustrates the selective retention capabilities of our method.

Comment #7:

Line 137-138: The difference in wettability of the 2PP-printed scaffolds can be attributed to the difference in surface tension of water and isopropanol plus the probably non-polar characteristic of the commercial resin. Are the contact angles in Supplementary Fig. 3 on the pure polymer film or modified with silane?

→We thank the reviewer for this question. All experiments in this work were conducted on the pure IPS scaffold without any surface modification. To make sure the same measurement conditions in all experiments, the contact angles in Supplementary Fig. 3 are measured on the polymer film without any surface treatment.

In the Supplementary Information:

Supplementary Fig. 7 Optical images showing the contact angle measurements of solutions containing 0.01 mM PBS solutions with different IPA/H₂O ratio on a IPS film (cross-linked) without any surface treatment.

Comment #8:

Line 183-186: How was the Au layer thickness measured?

→The technology of Focus Ion Beam (FIB) was used to cut the microstructure beam deposited with Au NPs, and then a cross-section SEM image of the beam was acquired. As shown in Supplementary Fig. S14, a clear interfacial contrast between the Au NP layer and the inner polymer scaffold in the SEM image was observed due to the huge difference in the conductivity of the Au NP layer and polymer beams. Then, the thickness of the Au NP layer can be measured by using the software of Fiji.

In the main text:

(Page 14, line 573) **Estimation of the thickness of the assembled Au NP layer.** The cross-section SEM image of a Au NP-assembled beam was first acquired after the FIB treatment. As shown in Supplementary Fig. 15, a clear interfacial contract between the assembled Au NP layer and the inner polymer scaffold was observed due to the huge difference in the conductivity of Au and polymer beam. The thickness of the assembled Au NP layer is determined by averaging thickness data extracted from different positions of the beam using the software of Fiji.

Comment #9:

Line 213-214: Did you also anneal the QD-coated structures and if yes, can the functionality of those be preserved?

→ We didn't anneal the quantum dots here, since their functionality would be decreased after annealing (Journal of Luminescence 177 (2016): 54-58). However, some approaches can be used to improve the thermal stability of QDs, for example, ligand exchanged, shell design, or inorganic overcoating (Advanced Materials 31.34 (2019): 1804294). Here, we mainly prove the possibility of depositing QDs onto various micro/nanostructures, showing the potential application of our method for creating various QDs micro/nanodevices. We have added some discussion in the revised conclusion section.

In the main text:

(Page 9, line 322) Furthermore, the exploration of milder etching technologies, such as plasma and wet etching, is necessary to extend the material applicability for fabricating pure-material microstructures (as some material would lose their functionality at high temperatures, e.g. QDs^{54,55}), improve the morphology and structural integrity of the pure-material micro/nanostructures.

REVIEWER COMMENTS

Reviewer #1 (Remarks to the Author):

The authors have addressed most of my questions and provided additional information. I have some additional comments as below.

1. In table 1, inorganic materials are considered to be embedded in the polymer matrix in ref 3 and 4. These two researches use nanomaterials as printing ink and no monomer exists during the printing process. Surface ligands capped on the nanocrystal surface are usually short alkane chains and they are not polymers. In fact, the nanomaterials used in this manuscript by Lyu et al. are also coated with various organic ligands. The authors should be more careful and objective when categorizing others' work and put their work in the right context.

2. Videos are needed for better understanding the failure behavior of the lattice structure during compression.

3. I also find the minimum linewidth of 400 nm to be somewhat contrived and some statistical characterization is needed. In addition, it seems that many unwanted nanoparticles deposited around the printed structures in SI fig.5. How much unintentional deposition around the printed structure is usually observed?

Reviewer #3 (Remarks to the Author):

the authors have appropriately addressed the majority of my questions and the manuscript is now clearer in those aspects.

There remains one open question that deserves maybe a better revision, linked to following new statement, as added by the authors:

(Page 12, line 499) In addition, a sacrificial layer could be applied prior to nanomaterial deposition for the scaffold recycles, and dissolving the layer could allow the assembled nanomaterials to release in the solution and thus recycle the scaffolds.

The question was related to the re-usability of the 3D printed, and costly scaffold. The proposed solution given by the authors is to add a sacrificial layer between the printed scaffold and the assembled particles. Such an approach would to my understanding limit severely the possible topology of the assembled structure, as the sacrificial layer etchant would need to access this film through openings in the assembled layer, and closed features become virtually impossible. It would be nice to address this opportunity and limits with a bit more arguments to better frame this idea. It would be great to have a proof-of-concept, but that would probably not be available at hand by the group to include here.

other than that point all my questions were answered. Well done job!

Reviewer #4 (Remarks to the Author):

Thanks for the careful review and answering to the comments. Any criticism was addressed properly and the article can be published in this state.

Response to Reviewers' comments

We appreciate the reviewers for their contribution on our manuscript, and have revised our manuscript based on the comments from the reviewers. Please find our point-by-point responses to the reviewers' comments preceded by blue arrows (→) below. Additionally, we have listed all changes made in either the main text or Supplementary Information files of our manuscript and highlighted them in yellow.

Reviewer #1:

The authors have addressed most of my questions and provided additional information. I have some additional comments as below.

→We thank the reviewer for the time and efforts on our manuscript.

Comment #1:

In table 1, inorganic materials are considered to be embedded in the polymer matrix in ref 3 and 4. These two researches use nanomaterials as printing ink and no monomer exists during the printing process. Surface ligands capped on the nanocrystal surface are usually short alkane chains and they are not polymers. In fact, the nanomaterials used in this manuscript by Lyu et al. are also coated with various organic ligands. The authors should be more careful and objective when categorizing others' work and put their work in the right context.

→We thank the reviewer for pointing out this error. We had miscategorized the distribution of desired material after printing for ref. 3,4 and ref. 5,6. We have accordingly corrected this in Supplementary Table 1.

Comment #2:

Videos are needed for better understanding the failure behavior of the lattice structure during compression.

→We thank the reviewer for this kind suggestion. We have included two videos showing the detailed failure behaviors of a 120 °C annealed microcube assembled with Au NPs (Supplementary video 3) and a 900 °C-annealed Fe₃O₄ sphere (Supplementary video 4) during compression.

In the main text:

(Page 7 line 249) The structural stability of our 3D microstructures can be significantly enhanced through thermal annealing (Fig. 4, Supplementary Video 3, 4).

Comment #3:

I also find the minimum linewidth of 400 nm to be somewhat contrived and some statistical characterization is needed.

→We thank the reviewer for raising this issue. We have re-calibrated the linewidth of 400 nm by averaging the width data extracted at different positions across the line. In addition, all data in Supplementary Fig. 5 are shown as mean \pm s.d. (n=6).

Comment #4:

In addition, it seems that many unwanted nanoparticles deposited around the printed structures in SI fig.5. How much unintentional deposition around the printed structure is usually observed?

→We thank the reviewer for raising this question. The number of nanoparticles deposited around the printed structure depends on several factors, such as the solution's affinity to the substrate, the concentration of nanomaterials, the trapped solution volume, and the number of immersing-retracting cycles. Due to the weak trapping capability of lines (only this specific case for resolution purpose in Supplementary Fig. 5), we used a relatively high Au NP concentration (~4 mg/mL) and a large solution volume (5 μ L) in each wetting and drying cycle, to make sure a high assembly content of Au NPs. For other 3D scaffolds, nanoparticles are primarily deposited onto the scaffolds instead of the substrates due to their selective affinity. We have added these details in the revised Supplementary Materials.

In the figure caption of Supplementary Fig.5:

Considering the weak trapping capability of lines, the experiment was conducted by dropping and drying 5 μ L 90% IPA/H₂O solution with ~4 mg/mL Au NP and 0.01 mM PBS for 12 cycles on the silane-treated substrate.

Reviewer #3:

The authors have appropriately addressed the majority of my questions and the manuscript is now clearer in those aspects. There remains one open question that deserves maybe a better revision, linked to following new statement, as added by the authors:

(Page 12, line 499) In addition, a sacrificial layer could be applied prior to nanomaterial deposition for the scaffold recycles, and dissolving the layer could allow the assembled nanomaterials to release in the solution and thus recycle the scaffolds.

The question was related to the re-usability of the 3D printed, and costly scaffold. The proposed solution given by the authors is to add a sacrificial layer between the printed scaffold and the assembled particles. Such an approach would to my understanding limit severely the possible topology of the assembled structure, as the sacrificial layer etchant would need to access this film through openings in the assembled layer, and closed features become virtually impossible. It would be nice to address this opportunity and limits with a bit more arguments to better frame this idea. It would be great to have a proof-of-concept, but that would probably not be available at hand by the group to include here.

→We thank the reviewer for their time and effort in reviewing our manuscript. Regarding the recycling of the templates, we agree with the reviewer that the etchant solutions need to access the film through openings in the assembled layer. In most cases, there are always gaps (even in closely packed nanospheres) or defects among the assembled nanoparticles, which allow the penetration of the etchant molecules (much smaller than these gaps) to react with the sacrificial layer. We have further elaborated on this idea and added more discussion in the revised manuscript.

In the main text:

(Page 12, line 499) In addition, a sacrificial layer could be applied prior to the nanomaterial deposition for the scaffolds recycle. **The etchant would penetrate through the gaps or defects among the assembled nanoparticles to dissolve the sacrificial layer, releasing the assembled nanomaterials into the solution, and the scaffolds are thus recycled.**

Other than that point all my questions were answered. Well done job!

→Thank you very much for the positive and valuable comments.

Reviewer #4:

Thanks for the careful review and answering to the comments. Any criticism was addressed properly and the article can be published in this state.

→We thank the reviewer for the time and efforts on our manuscript.

REVIEWER COMMENTS

Reviewer #1 (Remarks to the Author):

All my questions have been addressed. I can now recommend this paper for publication.

Reviewer #3 (Remarks to the Author):

In the revised version the suggestion to apply a sacrificial layer to reuse the scaffold has been more detailed, but it is not substantiated enough (e.g. what material and what etchant) and not demonstrated at all. The way to use pinholes and defects to remove the sacrificial layer may work but it is probably a very slow process and may create other issues, such as adding more defects, breaking of the thin shell, etc. At this point it is pure speculation and it is suggested not to mention it as it doesn't bring any value to the paper and merely proposes a possible implementation but without further details.

Response to Reviewers' comments

We appreciate the reviewers for their contribution on our manuscript, and have revised our manuscript based on the comments from Reviewer #3. Please find our point-by-point responses to the reviewers' comments preceded by blue arrows (→) below.

Reviewer #1:

All my questions have been addressed. I can now recommend this paper for publication.

→We thank the reviewer for the time and efforts on our manuscript.

Reviewer #3:

In the revised version the suggestion to apply a sacrificial layer to reuse the scaffold has been more detailed, but it is not substantiated enough (e.g. what material and what etchant) and not demonstrated at all. The way to use pinholes and defects to remove the sacrificial layer may work but it is probably a very slow process and may create other issues, such as adding more defects, breaking of the thin shell, etc. At this point it is pure speculation and it is suggested not to mention it as it doesn't bring any value to the paper and merely proposes a possible implementation but without further details.

→We thank the reviewer for their valuable suggestion. We agree that proposing this approach for template recycling is premature and does not contribute substantively to the paper. Therefore, we have deleted this description from the previous manuscript to ensure the rigor of our claims. Overall, we appreciate the reviewer's time and effort in reviewing our manuscript.